# Efficient First-Order Contextual Bandits:
## Prediction, Allocation, and Triangular Discrimination

**Dylan J. Foster**
Microsoft Research, New England
dylanfoster@microsoft.com

**Akshay Krishnamurthy**
Microsoft Research, NYC
akshaykr@microsoft.com

## Abstract

A recurring theme in statistical learning, online learning, and beyond is that faster convergence rates are possible for problems with low noise, often quantified by the performance of the best hypothesis; such results are known as *first-order* or *small-loss* guarantees. While first-order guarantees are relatively well understood in statistical and online learning, adapting to low noise in *contextual bandits* (and more broadly, decision making) presents major algorithmic challenges. In a COLT 2017 open problem, Agarwal et al. [5] asked whether first-order guarantees are even possible for contextual bandits and—if so—whether they can be attained by efficient algorithms. We give a resolution to this question by providing an optimal and efficient reduction from contextual bandits to online regression with the logarithmic (or, cross-entropy) loss. Our algorithm is simple and practical, readily accommodates rich function classes, and requires no distributional assumptions beyond realizability. In a large-scale empirical evaluation, we find that our approach typically outperforms comparable non-first-order methods.

On the technical side, we show that the logarithmic loss and an information-theoretic quantity called the *triangular discrimination* play a fundamental role in obtaining first-order guarantees, and we combine this observation with new refinements to the regression oracle reduction framework of Foster and Rakhlin [29]. The use of triangular discrimination yields novel results even for the classical statistical learning model, and we anticipate that it will find broader use.

## 1 Introduction

In the contextual bandit problem, a learning agent repeatedly makes decisions based on contextual information, with the goal of learning a decision-making policy that minimizes their total loss over time. This model captures simple reinforcement learning tasks in which the agent must learn to make high-quality decisions in an uncertain environment, but does not need to engage in long-term planning or credit assignment. Owing to the availability of high-quality engineered reward metrics, contextual bandit algorithms are now routinely deployed in production for online personalization systems [4, 61].

Contextual bandits encompass both the general problem of statistical learning with function approximation (specifically, cost-sensitive classification) and the classical multi-armed bandit problem, yet present algorithmic challenges greater than the sum of both parts. In spite of these difficulties, extensive research effort over the past decade has resulted in efficient, general-purpose algorithms, as well as a sharp understanding of the optimal worst-case sample complexity [9, 12, 3, 29, 58].

While the algorithmic and statistical foundations for contextual bandits are beginning to take shape, we still lack an understanding of *adaptive* or *data-dependent* algorithms that can go beyond the worst case and exploit nice properties of real-world instances for better performance. This is in stark contrast to supervised statistical learning, where adaptivity has substantial theory, and where standard

35th Conference on Neural Information Processing Systems (NeurIPS 2021).

algorithms (e.g., empirical risk minimization) are known to automatically adapt to nice data [17]. For contextual bandits, adaptivity poses new challenges that seem to require algorithmic innovation, and a major research frontier is to develop algorithmic principles for adaptivity and an understanding of the fundamental limits.

To highlight the lack of understanding for adaptive and data-dependent algorithms, a COLT 2017 open problem posed by Agarwal, Krishnamurthy, Langford, Luo, and Schapire [5] asks whether there exist contextual bandit algorithms that achieve a certain data-dependent *first-order* regret bound, which scales with the cumulative loss $L^\star$ of the best policy, rather than with the time horizon $T$. For multi-armed bandits, first-order regret bounds (also known as *small-loss bounds* or *fast rates*) typically scale as $\sqrt{L^\star}$ and imply faster convergence for "easy" problems, interpolating between the optimal $\sqrt{T}$ rate for worst-case instances and constant/logarithmic regret for noise-free instances [7, 31]. Agarwal et al. [5] observed that existing techniques appear to be inadequate to achieve this type of guarantee in contextual bandits. Beyond simply asking whether first-order regret can be achieved, they also asked whether it can be achieved *efficiently*, which is essential for real-world deployment. Subsequently, Allen-Zhu, Bubeck, and Li [6] gave an inefficient algorithm with an optimal first-order regret guarantee, resolving the former question, but the existence of efficient first-order algorithms remained open.

**Contributions.** We give the first optimal and efficient contextual bandit algorithm with a first-order regret guarantee, providing a resolution to the second open problem raised by Agarwal et al. [5]. Our algorithm, FastCB, builds on a recent line of research that develops efficient contextual bandit algorithms based on the computational primitive of (online/offline) *supervised regression* [43, 32, 29, 58], and is efficient in terms of queries to an *online oracle* for regression with the logarithmic loss. Beyond attaining first-order regret, FastCB inherits all of the benefits of recent algorithms based on regression: it is simple and practical, accommodates flexible function classes, requires no statistical assumptions beyond realizability, and enjoys strong empirical performance.

**Technical highlights.** By invoking the framework of regression oracles, our algorithm design approach deviates sharply from prior approaches to first-order regret and necessitates the use of techniques that are novel even in the context of statistical learning. At a high-level, the design of FastCB leverages two key techniques:

1. *First-order regret for plug-in classification via logarithmic loss:* We show that algorithms based on regression with least-squares, as used in prior work [29, 58, 68, 34, 23], fail to attain first-order regret, even for the simpler problem of cost-sensitive classification in statistical learning. In spite of this apparent setback, we show that regression with the logarithmic loss *does* lead to first-order regret for statistical learning. This is established through a new analysis based on an information-theoretic quantity called the *triangular discrimination* [66, 44, 62].

2. *Reweighted inverse gap weighting:* Moving from statistical learning to contextual bandits, we transform predictions into distributions over actions using a scale-sensitive refinement to the *inverse-gap weighting scheme* used in the SquareCB algorithm [1, 29]. Our new scheme is tailored to small losses, and we show that its error is controlled by the triangular discrimination.

Summarizing, our approach leverages **prediction** via the logarithmic loss, **allocation** via reweighted inverse gap weighting, and **triangular discrimination** as the bridge from prediction to allocation.

**Empirical results.** In Section 5, we evaluate FastCB on the large-scale contextual bandit benchmark of Bietti et al. [13] and find that it typically outperforms SquareCB and other non-adaptive baselines [35]. Interestingly, we observe that most of the performance improvement can be attributed to the use of the logarithmic loss, while the reweighted allocation scheme provides modest additional benefit. These findings raise a natural question as to whether simply moving to the logarithmic loss can yield performance improvements in production contextual bandit deployments.

**On the regression oracle model.** As a disclaimer, we caution that our algorithm is efficient in terms of an oracle for online regression, while Agarwal et al. [5] originally asked for an algorithm that is efficient in terms of a *cost-sensitive classification oracle* capable of solving the policy optimization problem $\operatorname{argmin}_{\pi \in \Pi} \sum_{t=1}^{T} \ell_t(\pi(x_t))$. Hence, while FastCB is the first algorithm with first-order regret that is efficient in *any* oracle model, it does not formally solve the original open problem. Nonetheless, there are strong reasons to prefer a solution based on regression over one based on classification. First, cost-sensitive classification is intractable to implement even for simple function classes for which regression can be solved efficiently [29]. Setting this issue aside, (online)

---
**Algorithm 1** FastCB ("Fast Rates for Contextual Bandits")
---
1: **parameters**:
      Learning rate $\gamma > 0$.
      Online regression oracle $\mathbf{Alg}_{\mathsf{KL}}$.
2: **for** $t = 1, \ldots, T$ **do**
3:       Receive context $x_t$.
         `// Compute oracle's predictions (Eq. (4)).`
4:       For each action $a \in \mathcal{A}$, compute $\widehat{y}_t(x_t, a) := \mathbf{Alg}_{\mathsf{KL}}^{(t)}(x_t, a \, ; \, \{(x_i, a_i, \ell_i(a_i))\}_{i=1}^{t-1})$.
5:       Let $b_t \in \operatorname{argmin}_{a \in \mathcal{A}} \widehat{y}_{t,a}$.
         `// Reweighted inverse gap weighting.`
6:       For each $a \neq b_t$, define $p_{t,a} = \frac{\widehat{y}_t(x_t, b_t)}{A\widehat{y}_t(x_t, b_t) + \gamma(\widehat{y}_t(x_t, a) - \widehat{y}_t(x_t, b_t))}$. Let $p_{t,b_t} = 1 - \sum_{a \neq b_t} p_{t,a}$.
7:       Sample $a_t \sim p_t$ and observe loss $\ell_t(a_t)$.
8:       Update $\mathbf{Alg}_{\mathsf{KL}}$ with example $(x_t, a_t, \ell_t(a_t))$.
9: **end for**
---

regression-based algorithms are typically simpler and faster than classification-based algorithms, and multiple empirical evaluations have shown that algorithms based on regression dominate those based on classification [32, 13, 35].

**Organization.**   Section 2 contains our algorithm and main theorem. Section 3 describes the motivation and analysis ideas behind FastCB, beginning from new techniques for statistical learning with regression-based classifiers. Examples for the main theorem are given in Section 4, and experimental results are given in Section 5. Detailed discussion of related work is deferred to Appendix A.

## 2   Main Result: An Efficient First-Order Algorithm for Contextual Bandits

We begin by formally introducing the contextual bandit model. At each round $t \in [T]$, the learner observes a context $x_t \in \mathcal{X}$, selects an action $a_t \in \mathcal{A}$, then observes a loss $\ell_t(a_t) \in [0,1]$ for the action they selected. We assume that $A := |\mathcal{A}|$ is finite and that each loss function $\ell_t : \mathcal{A} \to [0,1]$ is drawn independently from a fixed distribution $\mathbb{P}_{\ell_t}(\cdot \mid x_t)$, where $\mathbb{P}_{\ell_1}, \ldots, \mathbb{P}_{\ell_T}$ and $x_1, \ldots, x_T$ are selected by a potentially adaptive adversary.

We make a standard *realizability* assumption [24, 2, 32, 29]. Namely, we assume that the learner has access to a class of value functions $\mathcal{F} \subset (\mathcal{X} \times \mathcal{A} \to [0,1])$ (e.g., neural networks, kernels, or forests) that models the mean of the loss distribution.

**Assumption 1** (Realizability). *There exists $f^\star \in \mathcal{F}$ such that for all $t$, $f^\star(x, a) = \mathbb{E}[\ell_t(a) \mid x_t = x]$.*

The aim of the learner is to minimize their *regret* to the optimal policy $\pi^\star(x) := \operatorname{argmin}_{a \in \mathcal{A}} f^\star(x, a)$:

$$\mathbf{Reg}_{\mathsf{CB}}(T) := \sum_{t=1}^{T} \ell_t(a_t) - \sum_{t=1}^{T} \ell_t(\pi^\star(x_t)). \tag{1}$$

For each $f \in \mathcal{F}$, we let $\pi_f(x) := \operatorname{argmin}_{a \in \mathcal{A}} f(x, a)$ be the induced policy. We let $\Pi := \{\pi_f \mid f \in \mathcal{F}\}$ be the induced policy class.

**Further notation.**   We adopt standard big-oh notation, and write $f = \widetilde{\mathcal{O}}(g)$ to denote that $f = \mathcal{O}(g \max\{1, \operatorname{polylog}(g)\})$. We use $\lesssim$ only in informal statements to highlight the most salient elements of an inequality. We use $a \vee b = \max\{a, b\}$ and $a \wedge b = \min\{a, b\}$.

### 2.1   Algorithm and Main Result

FastCB builds on the SquareCB algorithm of Foster and Rakhlin [29], which provides an efficient, minimax-optimal reduction from contextual bandits to online regression with the square loss. Compared to SquareCB and other subsequent algorithms based on online regression [34, 23], the first twist here is that rather than working with the square loss, we build on the computational primitive of online regression with the *logarithmic loss*. While this point is inconsequential for worst-case guarantees, we show that it is a fundamental distinction for first-order guarantees.

**Online regression oracles.** In more detail, an online regression oracle, which we denote by $\mathbf{Alg}_{\mathsf{KL}}$ (for "Kullback-Leibler") operates in the following protocol: For each time $t$, the algorithm receives a context-action pair $(x_t, a_t)$, produces a prediction $\widehat{y}_t \in [0, 1]$, then receives a response $y_t$. The algorithm's prediction error is measured through the binary logarithmic/cross-entropy loss,

$$\ell_{\log}(\widehat{y}, y) := y \log(1/\widehat{y}) + (1 - y) \log(1/(1 - \widehat{y})). \tag{2}$$

The algorithm's goal is to ensure that the log loss regret to $\mathcal{F}$ is minimized for all sequences.

**Assumption 2.** *The algorithm $\mathbf{Alg}_{\mathsf{KL}}$ guarantees that for every (possibly adaptively chosen) sequence $x_{1:T}, a_{1:T}, y_{1:T}$, the log loss regret is bounded by a known function $\mathbf{Reg}_{\mathsf{KL}}(T)$:*

$$\sum_{t=1}^{T} \ell_{\log}(\widehat{y}_t, y_t) - \inf_{f \in \mathcal{F}} \sum_{t=1}^{T} \ell_{\log}(f(x_t, a_t), y_t) \leq \mathbf{Reg}_{\mathsf{KL}}(T), \tag{3}$$

Online regression with the logarithmic loss (or, *sequential probability assignment*) is a fundamental and well-studied problem in online learning, and there are efficient algorithms available for many function classes of interest [25, 67, 40, 37, 52, 55, 33, 48]; see Section 4 for examples. While log loss regret is a more stringent notion of performance than square loss regret, it nonetheless has a relatively mature theory characterizing optimal rates [57, 51, 20, 14].

**The algorithm.** FastCB (Algorithm 1) is a reduction that efficiently transforms any online regression oracle satisfying Assumption 2 into a contextual bandit algorithm with an optimal first-order regret bound. At each round $t$, the algorithm first computes the estimated loss

$$\widehat{y}_t(x_t, a) := \mathbf{Alg}_{\mathsf{KL}}^{(t)}(x_t, a \, ; \{(x_i, a_i, \ell_i(a_i))\}_{i=1}^{t-1}) \tag{4}$$

predicted by the regression oracle for each action $a$ (Line 4); see Appendix C.1 for a more detailed formal description of the oracle model. Next, FastCB uses these estimates to assign a probability of being played to each action $a$ via a scale-sensitive refinement to the inverse gap weighting strategy used in SquareCB [1, 29], which we call *reweighted inverse gap weighting* (Line 6). Letting $b_t := \operatorname{argmin}_{a \in \mathcal{A}} \widehat{y}_t(x_t, a)$ be the greedy action according to the predicted losses, we define

$$p_{t,a} := \frac{\widehat{y}_t(x_t, b_t)}{A\widehat{y}_t(x_t, b_t) + \gamma(\widehat{y}_t(x_t, a) - \widehat{y}_t(x_t, b_t))} \quad \forall a \neq b_t, \quad \text{and} \quad p_{t,b_t} := 1 - \sum_{a \neq b_t} p_{t,a}, \tag{5}$$

where $\gamma > 0$ is a learning rate parameter. Given this distribution, FastCB simply samples $a_t \sim p_t$, then updates the oracle with the resulting tuple $(x_t, a_t, \ell_t(a_t))$. Our main theorem shows that this leads to an optimal first-order regret bound.[1]

**Theorem 1** (Main theorem). *Suppose Assumptions 1 and 2 hold. Then Algorithm 1 guarantees that for all sequences with $\mathbb{E}\left[\sum_{t=1}^{T} \ell_t(\pi^\star(x_t))\right] \leq L^\star$, by choosing $\gamma = \sqrt{AL^\star/3\mathbf{Reg}_{\mathsf{KL}}(T)} \vee 10A$,*

$$\mathbb{E}[\mathbf{Reg}_{\mathsf{CB}}(T)] \leq 40\sqrt{L^\star \cdot A\mathbf{Reg}_{\mathsf{KL}}(T)} + 600A\mathbf{Reg}_{\mathsf{KL}}(T). \tag{6}$$

The dominant term in this regret bound scales with $\sqrt{L^\star}$ whenever the oracle $\mathbf{Alg}_{\mathsf{KL}}$ attains a fast $\log(T)$-type regret bound. As a simple example, whenever $\mathcal{F}$ is finite, we can instantiate $\mathbf{Alg}_{\mathsf{KL}}$ so that $\mathbf{Reg}_{\mathsf{KL}}(T) \leq \log|\mathcal{F}|$ [67], whereby FastCB enjoys optimal [2] first-order regret:

$$\mathbb{E}[\mathbf{Reg}_{\mathsf{CB}}(T)] \leq \mathcal{O}\left(\sqrt{L^\star \cdot A \log|\mathcal{F}|} + A \log|\mathcal{F}|\right).$$

Beyond first-order regret, FastCB inherits all of the advantages of online regression-based algorithms:

- *Efficiency and simplicity.* The memory and runtime used by the algorithm—on top of what is required by the regression oracle—scales only as $\mathcal{O}(A)$ per step; implementation is trivial.

- *Flexibility.* Working with regression as a primitive means that the algorithm easily accomodates rich, potentially nonparametric function classes, and we can instantiate Theorem 1 to get provable end-to-end regret guarantees for concrete classes of interest. For example, for linear models in $\mathbb{R}^d$ we can efficiently attain $\mathbf{Reg}_{\mathsf{KL}}(T) \leq \mathcal{O}(d \log(T))$ [25, 40], which yields a first-order regret bound $\mathbf{Reg}_{\mathsf{CB}}(T) \lesssim \sqrt{L^\star \cdot Ad}$; our result is new even for this simple special case. Similar guarantees are available for kernels, generalized linear models, and many other nonparametric classes. On the other hand, even for function classes where provable algorithms are not available, regression is amenable to practical heuristics (e.g., gradient descent). See Section 4 for detailed examples.

---

[1]While we assume that an upper bound on the optimal loss is known for simplicity, one can extend to the unknown case by running the algorithm in epochs, setting $\gamma$ in terms of the algorithm's estimated loss $L_t = \sum_{\tau=1}^{t} \ell_\tau(a_\tau)$, and applying the doubling trick. Theorem 1 also readily extends to high probability.

# 3 Overview of Analysis

We now outline the algorithmic principles and analysis ideas behind FastCB. First, in Section 3.1, we take a step back and consider the sub-problem of cost-sensitive classification in statistical learning. We establish that approaches based on least-squares fail to attain first-order regret (Theorem 2) for cost-sensitive classification, then show how to fix this problem using log loss regression (Theorem 3); this analysis serves as an introduction to the triangular discrimination. With this result in hand, we move to the contextual bandit setting and transform predictions into distributions over actions using the reweighted inverse-gap weighting scheme in (5), which exploits small losses. Our main result here shows that this scheme satisfies a first-order variant of the *per-round minimax inequality* of Foster and Rakhlin [29], which links the instantaneous contextual bandit regret to the triangular discrimination for the regression oracle on a per-round basis (Theorem 4). Full proofs are deferred to Appendices B and C.

## 3.1 Warmup: First-Order Regret Bounds for Plug-In Classifiers

For the simpler problem of cost-sensitive classification in statistical learning, the literature on *plug-in classification* shows that whenever realizability conditions such as Assumption 1 hold, we can obtain optimal worst-case regret by taking the greedy policy/classifier induced by a least-squares estimator. We first show that this approach fails to attain first-order regret.

The statistical learning setting we consider is as follows. We receive a dataset $D_n$ consisting of $n$ context-loss pairs $(x_t, \ell_t) \sim \mathcal{D}$ i.i.d., where the entire loss function $\ell_t : \mathcal{A} \to [0, 1]$ is observed. Analogously to Assumption 1, we assume access to a function class $\mathcal{F} \subseteq (\mathcal{X} \times \mathcal{A} \to [0, 1])$ such that $\mathbb{E}_{\mathcal{D}}[\ell(a) \mid x] = f^\star(x, a)$ for some $f^\star \in \mathcal{F}$, and take $\Pi := \{\pi_f \mid f \in \mathcal{F}\}$ as the induced class of policies. Our goal is to learn a policy $\widehat{\pi} : \mathcal{X} \to \mathcal{A}$ such that the regret (or, excess risk)

$$L(\widehat{\pi}) - L^\star \tag{7}$$

is small, where $L(\pi) := \mathbb{E}_{\mathcal{D}}[\ell(\pi(x))]$ and $L^\star := L(\pi^\star)$, with $\pi^\star := \pi_{f^\star}$. Formally, this an easier problem than contextual bandits, since any algorithm with a regret bound for contextual bandits yields a bound on the cost-sensitive classification regret (7) via online-to-batch conversion.

A classical result in statistical learning [65, 54, 59] shows that if we compute the policy/classifier $\widehat{\pi} := \operatorname{argmin}_{\pi \in \Pi} \sum_{t=1}^{n} \ell_t(\pi(x_t))$ that minimizes the empirical risk, we obtain a first-order regret bound of the form[2]

$$\mathbb{E}[L(\widehat{\pi})] - L^\star \lesssim \sqrt{\frac{L^\star \cdot \log|\mathcal{F}|}{n}} + \frac{\log|\mathcal{F}|}{n}. \tag{8}$$

This is an optimal first-order guarantee, but computing $\widehat{\pi}$ is typically computationally intractable, even for relatively simple policy classes. As an alternative, the approach of plug-in classification aims to use the realizability assumption to develop algorithms based on the more tractable primitive of regression. Here, another classical result (e.g., Audibert and Tsybakov [8][3]), shows that if we perform least-squares via

$$\widehat{f}_{\mathsf{LS}} := \operatorname*{argmin}_{f \in \mathcal{F}} \sum_{t=1}^{n} \sum_{a \in \mathcal{A}} (f(x_t, a) - \ell_t(a))^2,$$

and take $\widehat{\pi}_{\mathsf{LS}} := \pi_{\widehat{f}_{\mathsf{LS}}}$ as our classifier, then under the realizability assumption we are guaranteed

$$\mathbb{E}[L(\widehat{\pi}_{\mathsf{LS}})] - L^\star \lesssim \sqrt{\frac{A \log|\mathcal{F}|}{n}}. \tag{9}$$

While this result is rate-optimal, it is not first-order, and first-order regret bounds for plug-in classification are conspicuously absent from the literature. We show that this is fundamental.

**Theorem 2** (Failure of least-squares for plug-in classification)**.** *Let $\mathcal{A} = \{1, 2\}$ and $\mathcal{X} = \{1, 2\}$. For every $n > 10^8$, there exists a function class $\mathcal{F} \subseteq (\mathcal{X} \times \mathcal{A} \to [0, 1])$ with $|\mathcal{F}| = 2$, and a realizable distribution $\mathcal{D}$ such that $L^\star \leq \frac{2^7}{n} < 1$, yet $L(\widehat{\pi}_{\mathsf{LS}}) - L^\star \geq 2^{-5} \sqrt{\frac{1}{n}}$ with probability at least $1/10$.*

---

[2]Following the convention in contextual bandit literature, we focus on finite classes with $|\mathcal{F}| < \infty$ in this discussion, but one can extend our observations to general classes, e.g., using the machinery of Zhang [71].

[3]This result is well-known in the binary setting. We are not aware of a reference for the multiclass/cost-sensitive version here, though it is implicit in many recent works on contextual bandits.

Since the instance in this theorem has $\sqrt{\frac{L^\star \cdot A \log|\mathcal{F}|}{n}} \lesssim \frac{1}{n}$, we conclude that plug-in classification with least-squares fails to attain the first-order regret bound in (8) with constant probability; a lower bound in expectation follows immediately.

### 3.1.1   Fast Rates for Plug-In Classifiers: Triangular Discrimination and Logarithmic Loss

It would appear we are at an impasse, as Theorem 2 shows that square loss regression oracles of the type used in Foster and Rakhlin [29] are unlikely to attain first-order regret bounds on their own. However, the plug-in classification approach is not completely doomed. All we need to do to fix this issue is change the loss function and instead perform regression with the *logarithmic loss*.

To understand why plug-in least-squares fails and how it can be improved, it will be helpful to review the key steps in the analysis leading to the rate (9).

**Step 1**. First, using a generic regret decomposition based on realizability, for any $f$ we have

$$L(\pi_f) - L^\star \leq 2 \max_{\pi \in \{\pi_f, \pi^\star\}} \mathbb{E}_{\mathcal{D}} \big| f(x, \pi(x)) - f^\star(x, \pi(x)) \big|. \tag{10}$$

**Step 2**. Next, by Cauchy-Schwarz, for any policy $\pi$ we have

$$\mathbb{E}_{\mathcal{D}} \big| f(x, \pi(x)) - f^\star(x, \pi(x)) \big| \leq \left( \mathbb{E}_{\mathcal{D}} \big| f(x, \pi(x)) - f^\star(x, \pi(x)) \big|^2 \right)^{1/2}, \tag{11}$$

which we may further upper bound by $\left( \sum_{a \in \mathcal{A}} \mathbb{E}_{\mathcal{D}} \big| f(x, a) - f^\star(x, a) \big|^2 \right)^{1/2}$.

**Step 3**. Finally, under realizability, a standard concentration argument based on Bernstein's inequality implies that the least-squares estimator satisfies

$$\mathbb{E} \left[ \sum_{a \in \mathcal{A}} \mathbb{E}_{\mathcal{D}} \big| \widehat{f}_{\mathsf{LS}}(x, a) - f^\star(x, a) \big|^2 \right] \lesssim \frac{A \log|\mathcal{F}|}{n}. \tag{12}$$

Combining this bound with **Step 2**, we conclude that $\mathbb{E}[L(\widehat{\pi}_{\mathsf{LS}})] - L^\star \leq \sqrt{A \log|\mathcal{F}|/n}$.

The issue here is that even in the presence of low noise, the squared error in (12) shrinks no faster than $\frac{1}{n}$. This holds even if $L^\star \propto \frac{1}{n}$, as in the lower bound construction for Theorem 2. Consequently, once we apply Cauchy-Schwarz in **Step 2**, we lose all hope of attaining a first-order bound.

Our starting point toward improving this result is a refined application of Cauchy-Schwarz, by which we can replace the right hand side of (11) with

$$\left( \mathbb{E}_{\mathcal{D}}[f(x, \pi(x)) + f^\star(x, \pi(x))] \cdot \mathbb{E}_{\mathcal{D}} \left[ \frac{(f(x, \pi(x)) - f^\star(x, \pi(x)))^2}{f(x, \pi(x)) + f^\star(x, \pi(x))} \right] \right)^{1/2}. \tag{13}$$

The ratio term above is closely related to the *triangular discrimination*, an information-theoretic divergence measure which we define for $p, q \in \mathbb{R}_+^A$ as[4]

$$D_\Delta(p \,\|\, q) := \sum_a \frac{(p_a - q_a)^2}{p_a + q_a}. \tag{14}$$

The triangular discrimination—also known as the symmetric $\chi^2$-divergence and Vincze-Le Cam distance—is a fundamental, often-overlooked quantity in information theory [66, 44, 62]. Since readers may be unfamiliar, we record some basic facts.

**Proposition 1** (Topsøe [62])**.** *The triangular discrimination $D_\Delta$, over the domain $\Delta_A$, i) is the f-divergence given by $f(t) = \frac{(t-1)^2}{t+1}$, ii) is the square of a distance metric, and iii) is equivalent (up to a multiplicative constant) to both Hellinger distance and Jensen-Shannon divergence.*

---

[4]The triangular discrimination is traditionally defined over the simplex $\Delta_A$, but for our application it is useful to work with the entire positive orthant.

The triangular discrimination turns out to be "just right" for our purposes, in that it is both i) large enough to facilitate the scale-sensitive application of Cauchy-Schwarz in (13), and ii) small enough (compared to the more standard $\chi^2$-divergence) to facilitate minimizing from samples.

Returning to (13), we can upper bound with the triangular discrimination and leverage a certain *self-bounding* property that it satisfies to arrive at the following improvement on `Step 1`/`Step 2`.

**Lemma 1** (Regret decomposition for triangular discrimination). *For any* $f : \mathcal{X} \times \mathcal{A} \to [0, 1]$,

$$L(\pi_f) - L^\star \leq 8(L^\star \cdot \mathbb{E}_{\mathcal{D}}[D_\Delta(f^\star(x, \cdot) \| f(x, \cdot))])^{1/2} + 17\,\mathbb{E}_{\mathcal{D}}[D_\Delta(f^\star(x, \cdot) \| f(x, \cdot))]. \quad (15)$$

Lemma 1 shows that low triangular discrimination (i.e. $\mathbb{E}_{\mathcal{D}}[D_\Delta(f^\star(x, \cdot) \| f(x, \cdot))] \propto 1/n$) suffices for an optimal first-order regret bound. What remains is to find an estimator $\widehat{f}$ that minimizes this quantity given only samples. Our key observation here is that the triangular discrimination satisfies a refined variant of Pinsker's inequality (originally due to Topsøe [62]), which allows us to bound it by the Kullback-Leibler divergence:

$$D_\Delta(f^\star(x, \cdot) \| f(x, \cdot)) = \sum_a \frac{(f(x, a) - f^\star(x, a))^2}{f(x, a) + f^\star(x, a)} \leq 2 \sum_a d_{\mathrm{KL}}(f^\star(x, a) \| f(x, a)), \quad (16)$$

where $d_{\mathrm{KL}}(p \| q) := p \log(p/q) + (1 - p) \log((1 - p)/(1 - q))$ is the binary KL-divergence. Note that the triangular discrimination is critical here, as the *opposite* inequality holds for $\chi^2$-divergence. This bound suggests that we should minimize the logarithmic loss, since—under the realizability assumption—this loss is closely related to the KL-divergence. In particular, we show (Theorem 6 in Appendix B), that by taking the estimator

$$\widehat{f}_{\mathsf{KL}} := \underset{f \in \mathcal{F}}{\operatorname{argmin}} \sum_{t=1}^{n} \sum_{a \in \mathcal{A}} \ell_{\log}(f(x_t, a), \ell_t(a)),$$

we are guaranteed that with high probability, $\mathbb{E}_{\mathcal{D}}\big[D_\Delta(f^\star(x, \cdot) \| \widehat{f}_{\mathsf{KL}}(x, \cdot))\big] \lesssim \frac{A \log|\mathcal{F}|}{n}$. Putting everything together, we arrive at a first-order regret bound for the plug-in classifier $\widehat{\pi}_{\mathsf{KL}} := \pi_{\widehat{f}_{\mathsf{KL}}}$.[5]

**Theorem 3** (First-order regret bound for plug-in classification). *Let* $\delta \in (0, 1)$. *Suppose that Assumption 3 holds. Then with probability at least* $1 - \delta$, *we have*

$$L(\widehat{\pi}_{\mathsf{KL}}) - L^\star \leq 16\sqrt{\frac{L^\star \cdot A\left(\log|\mathcal{F}| + \log(A/\delta)\right)}{n}} + 68\frac{A\left(\log|\mathcal{F}| + \log(A/\delta)\right)}{n}.$$

Interestingly, applications of the triangular discrimination similar to Lemma 1 have recently been discovered across a number of branches of mathematics, including theoretical computer science (communication complexity lower bounds), probability, and group theory (e.g., construction of group homomorphisms) [70, 28, 11, 53]. Additionally, Bubeck and Sellke [18] use a related *non-negative $\chi^2$-divergence* to provide first-order Bayesian regret bounds for Thompson sampling for the multi-armed bandit.

### 3.2 Moving to Contextual Bandits: Inverse Gap Weighting meets Triangular Discrimination

FastCB builds on the development for plug-in classifiers in Section 3.1 but with two key differences. First, since we need to make decisions on the fly for arbitrary sequences of contexts, the algorithm estimates losses using an *online* regression oracle for the logarithmic loss, as described in Assumption 2. Second, and more importantly, since the algorithm receives partial feedback, the strategy for selecting actions is critical. Here our main technical result shows that the reweighted inverse gap weighting strategy (5) satisfies a certain *per-round* inequality that links the instantaneous contextual bandit error to the triangular discrimination between the oracle's prediction $\widehat{y}_t$ and the true loss function $f^\star$.

**Theorem 4** (First-order per-round inequality). *Let* $y \in [0, 1]^A$ *be given and* $b \in \operatorname{argmin}_a y_a$. *Define* $p_a = \frac{y_b}{Ay_b + \gamma(y_a - y_b)}$ *for* $a \neq b$, *and* $p_b = 1 - \sum_{a \neq b} p_a$. *If* $\gamma \geq 2A$, *then for all* $f \in [0, 1]^A$ *and* $a^\star \in \operatorname{argmin}_a f_a$, *we have*

$$\underbrace{\sum_a p_a(f_a - f_{a^\star})}_{\text{CB regret}} \leq \underbrace{\frac{5A}{\gamma} \sum_a p_a f_a}_{\text{bias from } \textit{exploring}} + \underbrace{7\gamma \sum_a p_a \frac{(y_a - f_a)^2}{y_a + f_a}}_{\text{error from } \textit{exploiting}}. \quad (17)$$

---

[5]The dependence on $A$ in this result can be improved under additional assumptions on the loss distribution. As an example, in Appendix B we remove the leading $A$ factor for the special case of multiclass classification.

The inequality (17) may be thought of as an algorithmic analogue of the refined Cauchy-Schwarz lemma (15), with the learning rate $\gamma$ modulating the tradeoff between exploration and exploitation. Applying the inequality for each step $t$ (with $p = p_t$, $y = \widehat{y}_t(x_t, \cdot)$, and $f = f^\star(x_t, \cdot)$), and using the Pinsker-type inequality (16), we are guaranteed that

$$\mathbb{E}[\mathbf{Reg}_{\mathsf{CB}}(T)] \leq \frac{5A}{\gamma}\mathbb{E}[L_T] + 14\gamma \cdot \mathbf{Reg}_{\mathsf{KL}}(T), \tag{18}$$

where $L_T := \sum_{t=1}^{T} \ell_t(a_t)$. By a standard argument, this implies the main result in Theorem 1.

Compared to the per-round inequality used to analyze the original version of SquareCB in Foster and Rakhlin [29], the main improvement given by Theorem 4 is that, by reweighting—which leads to less exploration when the optimal loss is small—we are able to replace a constant exploration bias of order $\frac{A}{\gamma}$ incurred by SquareCB with the scale-sensitive bias term $\frac{A}{\gamma} \cdot \sum_a p_a f_a$ in (17), leading to a first-order bound. The price for this improvement is that we must now minimize the triangular discrimination rather than the squared error used by SquareCB, but this is taken care of by the log loss oracle.

## 4 Examples

In this section we take advantage of the extensive literature on regression with the logarithmic loss [25, 67, 40, 37, 52, 55, 33, 48] and instantiate Theorem 1 to give provable and efficient first-order regret bounds for a number of function classes of interest. To the best of our knowledge, our results are new for each of these special cases.

**Example 1** (Finite function classes). *If $\mathcal{F}$ is a finite class, Vovk's aggregating algorithm [67] guarantees that[6] $\mathbf{Reg}_{\mathsf{KL}}(T) \leq \log|\mathcal{F}|$. With this choice, FastCB satisfies $\mathbb{E}[\mathbf{Reg}_{\mathsf{CB}}(T)] \leq \mathcal{O}\big(\sqrt{L^\star \cdot A\log|\mathcal{F}|} + A\log|\mathcal{F}|\big)$.*

**Example 2** (Low-dimensional linear functions). *Suppose that $\mathcal{F}$ takes the form $\mathcal{F} = \{(x, a) \mapsto \langle w, \phi(x, a)\rangle \mid w \in \Delta_d\}$, where $\phi(x, a) \in \mathbb{R}_+^d$ is a fixed feature map with $\|\phi(x, a)\|_\infty \leq 1$. Then the continuous exponential weights algorithm ensures that $\mathbf{Reg}_{\mathsf{KL}}(T) \leq \mathcal{O}(d\log(T/d))$, and can be implemented in $\mathrm{poly}(d, T)$ time per step using log-concave sampling [25, 40].[7] With this choice, FastCB satisfies $\mathbb{E}[\mathbf{Reg}_{\mathsf{CB}}(T)] \leq \mathcal{O}\big(\sqrt{L^\star \cdot Ad\log(T/d)} + Ad\log(T/d)\big)$.*

Beyond attaining first-order regret, the bound in this example is minimax optimal when the number of actions is constant [46]. A natural direction for future work is to extend the result to large action spaces. Another more practical choice for the oracle in this setting is the algorithm of Luo et al. [48], which has slightly worse regret $\mathbf{Reg}_{\mathsf{KL}}(T) \leq \widetilde{\mathcal{O}}(d^2)$, but runs in time $\mathcal{O}(Td^{2.5})$ per step.

While first-order regret bounds for contextual bandits have primarily been investigated for finite classes prior to this work, an advantage of working within the regression oracle framework is that we can easily lift our first-order guarantees to rich, nonparametric function classes.

**Example 3** (High/infinite-dimensional linear functions). *Suppose that $\mathcal{F}$ takes the form $\mathcal{F} = \{(x, a) \mapsto \frac{1}{2}(1 + \langle w, \phi(x, a)\rangle) \mid \|w\|_2 \leq 1\}$, where $\|\phi(x, a)\|_2 \leq 1$ is a fixed feature map. For this setting, Rakhlin and Sridharan [55, Section 6.1] show that the FTRL algorithm with log-barrier regularization has[8] $\mathbf{Reg}_{\mathsf{KL}}(T) \leq \mathcal{O}(\sqrt{T\log(T)})$. This algorithm can be implemented in time $\mathcal{O}(d)$ per step, and satisfies the dimension-independent rate $\mathbb{E}[\mathbf{Reg}_{\mathsf{CB}}(T)] \leq \mathcal{O}\big((AL^\star)^{1/2}T^{1/4} + A\sqrt{T}\big)$.*

Let us interpret the bound in Example 3. First, we recall that the minimax optimal rate for this setting is $A^{1/2}T^{3/4}$, which the bound above always achieves in the worst case [1, 29]; this "worse-than-$\sqrt{T}$" rate is the price we pay for working with an expressive function class. On the other hand, if $L^\star$ is constant, the bound in Example 3 improves to $\mathcal{O}(A\sqrt{T})$, which beats the worst-case rate. While one might hope that a tighter rate of the form, e.g., $(L^\star)^{3/4}$, might be possible, by adapting a lower bound in Srebro et al. [59, Section 4], one can show that this result cannot be improved.

Lastly, we highlight that the logarithmic loss is well-suited to generalized linear models.

---

[6]See Proposition 6 for a proof that the loss $\ell_{\log}(\widehat{y}, y)$ is mixable over the domain $[0, 1]$.

[7]Our setup directly reduces to universal portfolio selection as follows: When $y_t$ is binary, we reduce by using features $\phi(x_t, a_t)$ when $y_t = 1$, and using features $\mathbf{1}_d - \phi(x_t, a_t)$ when $y_t = 0$. The case where $y_t \in [0, 1]$ can be reduced to this setting by sampling from $\mathrm{Ber}(y_t)$.

[8]This is technically only proven for the case where $y \in \{0, 1\}$, but the proof easily extends to $y \in [0, 1]$.

**Example 4** (Generalized linear models). *Let* $\mathcal{F} = \left\{ (x, a) \mapsto \sigma(\langle w, \phi(x, a) \rangle) \mid w \in \mathbb{R}^d, \|w\|_2 \leq 1 \right\}$, *where* $\sigma(t) = 1/(1 + e^{-t})$ *is the logistic link function and* $\phi(x, a)$ *is a fixed feature map. In this case, the map* $w \mapsto \ell_{\log}(\sigma(\langle w, \phi(x, a) \rangle), y)$ *is equivalent to the standard logistic loss function applied to* $\langle w, \phi(x, a) \rangle$, *and we can use the algorithm from Foster et al. [33] to obtain* $\mathbf{Reg}_{\mathsf{KL}}(T) \leq \mathcal{O}(d \log(T/d))$ *and* $\mathbf{Reg}_{\mathsf{CB}}(T) \leq \widetilde{\mathcal{O}}(\sqrt{L^\star \cdot Ad} + Ad)$.

Beyond the algorithmic examples above, for general function classes Bilodeau et al. [14] provide a tight characterization for the minimax optimal rates for online regression with the logarithmic loss in terms of *sequential covering numbers* [55] for the class $\mathcal{F}$. We can use this result in tandem with Theorem 1 to give new regret bounds for general classes.

## 5 Experiments

We compared the performance of FastCB to that of the de-facto alternative, SquareCB [29] in the large-scale contextual bandit evaluation suite ("bake-off") of Bietti et al. [13]. We found that FastCB typically enjoys improved performance, particularly on datasets where the optimal loss $L^\star$ is small. As a secondary observation, we found that using generalized linear models with the logarithmic loss rather than a linear model with the square loss (as in prior work [13, 35]) leads to substantial improvements, even without changing the SquareCB allocation rule. We summarize results here; further details are given in Appendix E.

**Datasets.** The *contextual bandit bake-off* is a collection of over 500 multiclass, multilabel, and cost-sensitive classification datasets available on the openml.org platform [64]. The collection was introduced in Bietti et al. [13] for the purpose of benchmarking oracle-based contextual bandit algorithms. Following Bietti et al. [13], we use the multiclass classification datasets from the collection (each context $x$ has a "correct" label $y$ associated with it) to simulate bandit feedback by assigning loss 0 if the learner predicts the correct label and 1 otherwise.

**Algorithms and oracle.** We use the standard implementation of SquareCB in the Vowpal Wabbit (VW) online learning library,[9] as used by Foster et al. [35]. We also implement FastCB in VW.

For both algorithms, we instantiate the oracle as performing online logistic regression with a fixed dataset-dependent feature map. This choice is convenient because i) it naturally produces predictions in $[0, 1]$, as required by FastCB, and ii), it formally meets our oracle requirements, since it is equivalent to online log loss regression with a generalized linear model. It can also be viewed as an admissible online square loss oracle, as required by SquareCB (see Appendix E for further discussion). We additionally instantiate SquareCB with a linear model and the square loss, which was shown to be the strongest non-adaptive method in prior evaluations [35]. We do not compare with high-performing adaptive algorithms like RegCB and AdaCB [13, 35] as these algorithmic modifications are somewhat complementary, and we expect they can be incorporated into FastCB. All oracles are trained with the default VW learning rule, which performs online gradient descent with adaptive updates [27, 41, 56].

For both FastCB and SquareCB, we apply inverse gap weighting (the reweighted and original version, respectively) with a time-varying learning rate schedule in which we set $\gamma = \gamma_t$ in Line 6 of Algorithm 1 at round $t$, and likewise for SquareCB. Following Foster et al. [35], we set $\gamma_t = \gamma_0 t^\rho$, where $\gamma_0 \in \{10, 50, 100, 400, 700, 10^3\}$ and $\rho \in \{.25, .5\}$ are hyperparameters.

**Evaluation.** We evaluate the performance of each algorithm using *progressive validation* (PV) loss, defined as $L_{\mathsf{PV}}(T) = \frac{1}{T} \sum_{t=1}^T \ell_t(a_t)$ [15]. Following Bietti et al. [13], we define a given algorithm as beating another algorithm *significantly* on a given dataset using an approximate $Z$-test. For each pair $(a, b)$ of algorithms, Figure 1 (top row) displays the number of datasets where $a$ beats $b$ significantly, minus the number of datasets where $b$ beats $a$ significantly. Figure 1 (bottom row) shows the progressive validation loss for the best-performing hyperparameter configuration for each algorithm as a function of the number of examples. We consider 10 replicates for each dataset, where each replicate has the example order randomly permuted, and plot the average progressive validation loss across the replicates. Error bands in each plot correspond to significance $p < 0.05$ under the $Z$-test (cf. (41)). See Appendix E for details.

**Results.** We find (Figure 1, top row) that FastCB with the logistic loss oracle (FastCB.L) has a positive win-loss difference against SquareCB with both logistic and square loss oracles

---

[9] https://vowpalwabbit.org

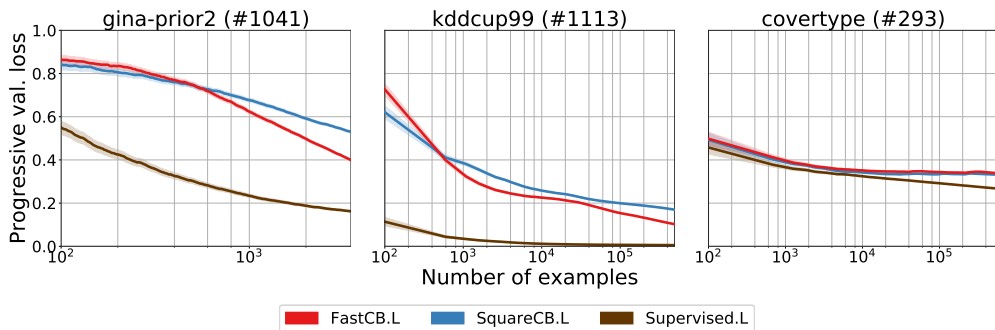

| ↓ vs → | S.S | S.L | F.L |
|---|---|---|---|
| SquareCB.S | - | -55 | -66 |
| SquareCB.L | 55 | - | -11 |
| **FastCB.L** | **66** | **11** | - |

| ↓ vs → | S.S | S.L | F.L |
|---|---|---|---|
| SquareCB.S | - | -54 | -64 |
| SquareCB.L | 54 | - | -3 |
| **FastCB.L** | **64** | **3** | - |

Figure 1: *Top:* Head-to-head win-loss differences. Each entry indicates the statistically significant win-loss difference between the row algorithm and the column algorithm. *Top-Left:* All hyperparameters are optimized on each dataset. *Top-Right:* Best fixed hyperparameter configuration across all datasets; only the oracle's learning rate is optimized per-dataset. *Bottom:* Progressive validation results for representative datasets depicting significant wins for FastCB.L (left, center) and a loss (right).

(SquareCB.L/SquareCB.S), indicating the strongest overall performance. This holds both when hyper-parameters are optimized on a per-dataset basis and for the best global hyperparameter configuration.

Perhaps surprisingly, our results suggest that the largest gains come from switching from the square loss oracle to the logistic loss oracle (SquareCB.S vs. SquareCB.L), while the gains from switching from the original inverse gap weighting strategy to our reweighted version (SquareCB.L vs. FastCB.L) are more marginal. Inspecting the results in more detail, we find that when we compare FastCB.L and SquareCB.L with hyperparameters optimized on a per-dataset basis, FastCB.L wins on 14/17 of the datasets in which either algorithm wins significantly, and that all but two of these 14 datasets have $L^\star \leq 0.2$. This suggests that the reweighted inverse gap weighting strategy is indeed helpful when $L^\star$ is small. Figure 1 (bottom row) displays progressive validation performance for FastCB.L and SquareCB.L for three representative datasets which illustrate this phenomenon.

The fact that FastCB.L does not strictly improve over SquareCB.L on every dataset, in spite of being very similar, might be attributed to the fact that the constants in the per-round inequality (17) are worse than those in the corresponding inequality for SquareCB.L, suggesting worse performance when $L^\star$ is not small. Thus, a fruitful future direction might be to find a strategy with optimal constants for (17).

## 6 Discussion

We have given the first efficient algorithm with optimal first-order regret for contextual bandits, resolving a variant of the open problem posed by Agarwal et al. [5]. Let us briefly mention some extensions. First, we believe that our techniques can also be used to obtain first-order guarantees for stochastic contextual bandits with an *offline* log loss oracle (à la Simchi-Levi and Xu [58])—albeit with a more technical analysis. As another extension, in Appendix D we show how to use our method to efficiently obtain a first-order regret bound when working with rewards rather than losses. Such a guarantee is useful when no policy accumulates much reward, as is common in personalization applications. Several other extensions appear to be straightforward, including accommodating infinite action spaces [34].

We close with some directions for future work. Directly relevant to our theoretical results is to continue the investigation into adaptivity in contextual bandits and reinforcement learning. More broadly, while triangular discrimination has been used in various mathematics disciplines, we are not aware of many applications in algorithm design. Are there other uses for the triangular discrimination in machine learning? We look forward to pursuing these directions.

## Acknowledgments and Disclosure of Funding

We thank Sivaraman Balakrishnan, John Langford, Zakaria Mhammedi, and Sasha Rakhlin for helpful discussions. We also thank Sasha Rakhlin for providing Google Cloud credits used for experiments.

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
