# A  Further Related Work

First-order regret bounds have a long history throughout statistical learning [65, 54, 59], online learning [36, 10, 21, 22, 47, 42, 30], and bandits [7, 31, 5, 49, 6]. Below we highlight some of the most relevant lines of work.

**Statistical learning and plug-in classification.**    Beginning with the work of Vapnik and Chervonenkis [65] for VC classes, classical work in statistical learning [54, 59] provides first-order regret (or, excess risk) bounds for *empirical risk minimization* which, in our setting, corresponds to the (typically intractable) policy optimization problem $\mathrm{argmin}_{\pi \in \Pi} \sum_{t=1}^{T} \ell_t(\pi(x_t))$. These results are also sometimes referred to as relative deviation bounds.

In the realizable setting (i.e., under Assumption 1), the process of fitting a model $\widehat{f}$ for the losses using regression and then performing classification with the induced classifier $\pi_{\widehat{f}}$ is often referred to as *plug-in classification* [69, 8, 26]. While these works establish worst-case optimal guarantees for plug-in classifiers, first-order regret bounds are—to the best of our knowledge—unexplored, and our observations regarding the suboptimality of least-squares and optimality of log loss regression are new.

**Bandits.**    First-order regret bounds for multi-armed bandits appear in Allenberg et al. [7] (see also Foster et al. [31], Bubeck and Sellke [18]), and have been extended to the semi-bandit framework [50, 49] and linear bandits [39]. For contextual bandits, Agarwal et al. [5] show that many common algorithms fall short of achieving first-order regret, and we are not aware of any optimal first-order algorithms outside the solution of Allen-Zhu et al. [6], even if one disregards efficiency or considers additional assumptions such as realizability.

On the technical side, Bubeck and Sellke [18] provide first-order regret bounds for Thompson sampling for the multi-armed bandit in the Bayesian setting. Their approach takes advantage of a certain *nonnegative $\chi^2$-divergence* which is closely related to the triangular discrimination we work with. Curiously, their analysis uses this divergence to measure distance between (posterior) distributions over actions, whereas we use the triangular discrimination to measure distance between regression functions. It would be interesting to understand whether there are deeper (e.g., primal-dual) connections between these approaches.

**Fast rates under margin/gap conditions.**    Another line of work on plug-in classifiers aims for faster rates under various margin assumptions, and—similar to our work—observes that least-squares can be suboptimal in certain settings [8]. Fast rates based on margin conditions are distinct from first-order bounds (neither type of bound implies the other in general), but it would be interesting to understand their relationship more closely. Recent work [35] extends these developments to contextual bandits and provides logarithmic regret bounds based on similar gap/margin conditions. As in statistical learning, these types of guarantees are incomparable to first-order regret bounds.

**Heteroscedastic regression.**    Our observations regarding suboptimality of least-squares for plug-in classification are also closely related to regression with heteroscedastic noise (Carroll [19]; Takeshi [60, Chapter 6]). Consider a regression setting where we receive variables $\{(x_i, y_i)\}_{i=1}^{n}$ i.i.d., with $y_i = f^\star(x_i) + \varepsilon_i$ for some $f^\star \in \mathcal{F}$, where $\mathbb{E}[\varepsilon_i \mid x_i] = 0$, and our goal is to produce an estimator such that the $L_1$-error $\mathbb{E}|\widehat{f}(x) - f^\star(x)|$ is small. In the heteroscedastic model, the noise variance $\sigma_x^2 := \mathbb{E}[\varepsilon_i^2 \mid x_i = x]$ may vary as a function of $x$. Using the same construction as Theorem 2, one can show that standard least-squares incurs error scaling with the worst-case variance $\sup_x \sigma_x^2$, while, if the variances were known, weighted least-squares with weights $w_x := 1/\sigma_x^2$ would yield error scaling with the more favorable average variance $\mathbb{E}[\sigma_x^2]$. Key to our results is that for responses in $[0, B]$, we have $\mathbb{E}[\sigma_x^2] \leq B \cdot \mathbb{E}[f^\star(x)]$ and, as we show, the logarithmic loss achieves error scaling with the latter quantity *without knowledge of the variances*. We mention in passing that regression with heteroscedastic noise has found recent use in the context of reinforcement learning with linear function approximation [73, 72].

# B  Proofs for Plug-In Classification Results (Section 3.1)

## B.1  Proof of Theorem 2

**Theorem 2** (Failure of least-squares for plug-in classification). *Let $\mathcal{A} = \{1, 2\}$ and $\mathcal{X} = \{1, 2\}$. For every $n > 10^8$, there exists a function class $\mathcal{F} \subseteq (\mathcal{X} \times \mathcal{A} \to [0, 1])$ with $|\mathcal{F}| = 2$, and a realizable distribution $\mathcal{D}$ such that $L^\star \leq \frac{2^7}{n} < 1$, yet $L(\widehat{\pi}_{\mathsf{LS}}) - L^\star \geq 2^{-5}\sqrt{\frac{1}{n}}$ with probability at least $1/10$.*

**Proof.** Let $\widehat{L}_{\mathsf{LS}}(f) = \frac{1}{n}\sum_{t=1}^n \sum_{a \in \mathcal{A}}(f(x_t, a) - \ell_t(a))^2$ be the empirical square loss, so that $\widehat{f}_{\mathsf{LS}} = \operatorname{argmin}_{f \in \mathcal{F}} \widehat{L}_{\mathsf{LS}}(f)$. We adopt the shorthand $\varepsilon_n = 1/n$ throughout the proof.

**Construction.**  We define $\mathcal{X} = \{x^{(1)}, x^{(2)}\}$ and $\mathcal{A} = \{a^{(1)}, a^{(2)}\}$, so that there are only two possible contexts and actions.

The data-generating process for our construction has three parameters, $\mu_n$, $\nu_n$, and $p_n$. We choose $\mathbb{P}_{\mathcal{D}}(x = x^{(1)}) = 1 - p_n$, and define $f^\star$ and the conditional loss distribution as follows:

- $f^\star(x^{(1)}, a^{(1)}) = \mu_n$ and $f^\star(x^{(1)}, a^{(2)}) = \nu_n$, where $\mu_n < \nu_n$. We choose $\ell(a^{(1)}) \sim \operatorname{Ber}(\mu_n) \mid x^{(1)}$ and $\ell(a^{(2)}) = \nu_n$ a.s. $\mid x^{(1)}$.
- $f^\star(x^{(2)}, a^{(1)}) = f^\star(x^{(2)}, a^{(2)}) = \frac{1}{2}$. We choose $\ell(a^{(1)}) \sim \operatorname{Ber}(\frac{1}{2}) \mid x^{(2)}$ and $\ell(a^{(2)}) = \frac{1}{2}$ a.s. $\mid a^{(2)}$.

We take $\mathcal{F} = \{f^\star, \tilde{f}\}$, where $\tilde{f}$ will be fully specified in the sequel, but is chosen to satisfy $\tilde{f}(x, a^{(2)}) = f^\star(x, a^{(2)})$ for all $x$. This, combined with the fact that $\ell(a^{(2)})$ is deterministic conditioned on $x$, means that our analysis will only concern the realized outcomes for $\ell(a^{(1)})$.

The high level idea for our construction is to set $p_n, \mu_n \propto \varepsilon_n = 1/n$, which ensures that $L^\star \leq (1 - p_n)\mu_n + p_n \lesssim \frac{1}{n}$, then show that if we choose $\tilde{f}(\cdot, a^{(1)}) \approx (\sqrt{\varepsilon_n}, 0)$, we have $\widehat{f}_{\mathsf{LS}} = \tilde{f}$ with constant probability. We then choose $\nu_n \approx \sqrt{\varepsilon_n}/2$, which implies that $\pi_{\tilde{f}}(x^{(1)}) = a^{(2)} \neq \pi^\star(x^{(1)})$, and consequently

$$L(\widehat{\pi}_{\mathsf{LS}}) = L(\pi_{\tilde{f}}) \gtrsim (1 - p_n) \cdot f^\star(x^{(1)}, \pi_{\tilde{f}}(x^{(1)})) = (1 - p_n) \cdot \nu_n \gtrsim \sqrt{\varepsilon_n}.$$

We make this approach formal below.

**Bad event.**  Let $n_1$ and $n_2$ be the number of examples for which $x = x^{(1)}$ and $x = x^{(2)}$. Let $n_1(0)$ and $n_1(1)$ be the number of examples for which $x = x^{(1)}$ and $\ell(a^{(1)}) = 0$ or $\ell(a^{(1)}) = 1$, respectively, and let $n_2(0)$ and $n_2(1)$ be defined likewise. We restrict to $n \geq 4$ going forward so that $\varepsilon_n \leq 1/4$.

Let $\widehat{\mu}_1 = \frac{1}{n_1}\sum_{i:x_i = x^{(1)}} \ell(a^{(1)})$ (whenever $n_1 > 0$), and let $\widehat{\mu}_2$ be defined likewise.

We prove the following proposition, which states that a certain event that is unfavorable for the least-squares estimator occurs with constant probability.

**Proposition 2.** *Let $n \geq 256$. Then if we set $p_n = \varepsilon_n$ and $\mu_n = 2^7\varepsilon_n$, the following event holds with probability at least $1/10$.*

1. *$n_2 = n_2(0) = 1$, and in particular $\widehat{\mu}_2 = 0$.*
2. *$n_1 \geq \frac{3}{8}n$.*
3. *$\widehat{\mu}_1 \leq \frac{3}{2}\mu_n$.*

Going forward, we adopt the parameter setting in Proposition 2 and condition on the event in the proposition, which we denote by $\mathscr{E}$. Note that this parameter setting ensures that

$$L^\star = (1 - \varepsilon_n)f^\star(x^{(1)}, a^{(1)}) + \varepsilon_n f^\star(x^{(2)}, a^{(1)}) = (1 - \varepsilon_n)\mu_n + \frac{\varepsilon_n}{2} \leq 2^8\varepsilon_n,$$

as long as $\mu_n < \nu_n$.

**Lower bound under the bad event.** Next, we observe that for both $f \in \mathcal{F}$, since $f(x, a^{(2)})$ perfectly predicts $\ell(a^{(2)})$ for all $x$, we have

$$\widehat{L}_{\mathsf{LS}}(f) \equiv \frac{n_1}{n}(f(x^{(1)}, a^{(1)}) - \widehat{\mu}_1)^2 + \frac{n_2}{n}(f(x^{(2)}, a^{(1)}) - \widehat{\mu}_2)^2,$$

up to additive noise that depends only on the realization of the dataset, not on the function $f$ under consideration. Since our argument only depends on the relative value of $\widehat{L}_{\mathsf{LS}}$, we identify $\widehat{L}_{\mathsf{LS}}$ with this representation going forward. We first observe that conditioned by Proposition 2 (Item 1), we have $\widehat{\mu}_2 = 0$, so that

$$\widehat{L}_{\mathsf{LS}}(f^\star) \geq \frac{n_2}{n}(f^\star(x^{(2)}, a^{(1)}) - \widehat{\mu}_2)^2 = \varepsilon_n \cdot (f^\star(x^{(2)}, a^{(1)}))^2 = \frac{\varepsilon_n}{4}.$$

Here we use that $n_2 = 1$ under the bad event and that $\varepsilon_n = 1/n$. On the other hand, if we set $\tilde{f}(x^{(2)}, a^{(1)}) = 0$, we have

$$\begin{aligned}
\widehat{L}_{\mathsf{LS}}(\tilde{f}) = \frac{n_1}{n}(\tilde{f}(x^{(1)}, a^{(1)}) - \widehat{\mu}_1)^2 &\leq 2(\tilde{f}(x^{(1)}, a^{(1)}))^2 + 2\widehat{\mu}_1^2 \\
&\leq 2(\tilde{f}(x^{(1)}, a^{(1)}))^2 + 2^3 \mu_n^2 \\
&\leq 2(\tilde{f}(x^{(1)}, a^{(1)}))^2 + 2^{17}\varepsilon_n^2,
\end{aligned}$$

where we have used Proposition 2 (Item 3). Note that as long as $\varepsilon_n < 2^{-20}$, we have $2^{17}\varepsilon_n^2 < \varepsilon_n/8$. If this is satisfied, then by choosing $\tilde{f}(x^{(1)}, a^{(1)}) = \sqrt{\varepsilon_n/16}$, we have

$$\widehat{L}_{\mathsf{LS}}(\tilde{f}) < \frac{\varepsilon_n}{4} \leq \widehat{L}_{\mathsf{LS}}(f^\star),$$

and we conclude that $\widehat{f}_{\mathsf{LS}} = \tilde{f} \neq f^\star$ whenever $\mathscr{E}$ occurs.

To conclude, we set $\nu_n = \sqrt{\varepsilon_n}/8$. Since $\tilde{f}(x^{(1)}, a^{(2)}) = \nu_n$, we have $\tilde{f}(x^{(1)}, a^{(2)}) < \tilde{f}(x^{(1)}, a^{(1)})$, so that $\pi_{\tilde{f}}(x^{(1)}) = a^{(2)} \neq \pi^\star(x^{(1)})$; this choice satisfies $\mu_n < \nu_n$ as required as long as $\varepsilon_n < 2^{-20}$. Finally, we observe that

$$L(\pi_{\tilde{f}}) - L^\star = (1 - \varepsilon_n)(\nu_n - \mu_n) \geq \frac{1}{2}(\sqrt{\varepsilon_n}/8 - 2^7 \varepsilon_n) > 2^{-5}\sqrt{\varepsilon_n},$$

as long as $\varepsilon_n < 2^{-22}$. $\qquad\square$

**Proof of Proposition 2.** Let $\mathscr{E}_1$, $\mathscr{E}_2$, and $\mathscr{E}_3$ denote the respective events in Proposition 2. We lower bound their probabilities one by one.

**Event $\mathscr{E}_1$.** We calculate

$$\mathbb{P}(n_2 = 1) = \sum_{i=1}^n \varepsilon_n \cdot (1 - \varepsilon_n)^{n-1} = \frac{1}{1 - \varepsilon_n}(1 - \varepsilon_n)^{1/\varepsilon_n} \geq e^{-1},$$

where we have used that $(1 - 1/x)^x \geq e^{-1}(1 - 1/x)$ for $x \geq 1$. Hence, since $\ell(a^{(1)}) \sim \mathrm{Ber}(\frac{1}{2})$ given $x^{(2)}$, $\mathscr{E}_1$ happens with probability at least $1 - \delta_1$ for $\delta_1 := 1 - e^{-1}/2$.

**Event $\mathscr{E}_2$.** We recall a standard multiplicative variant of the Chernoff bound.

**Lemma 2** (Chernoff bound (e.g., Boucheron et al. [16])). *Let $Y_i \sim \mathrm{Ber}(\mu)$ i.i.d.. Then for any* $x \in [0, 1/2]$,

$$\mathbb{P}\left(\sum_{i=1}^n Y_i \geq (1 + x)\mu n\right) \vee \mathbb{P}\left(\sum_{i=1}^n Y_i \leq (1 - x)\mu n\right) \leq e^{-\frac{1}{4}x^2 \mu n}.$$

As long as $p_n = 1/n \leq 1/4$, Lemma 2 implies that $n_1 \geq \frac{3n}{8}$ with probability at least $1 - e^{-\frac{3n}{64}} =: 1 - \delta_2$, so that event $\mathscr{E}_2$ holds.

**Event $\mathscr{E}_3$.** We observe that conditioned on the realization of $x_1, \ldots, x_n$, Lemma 2 implies that

$$\widehat{\mu}_1 \leq \frac{3}{2}\mu_n$$

with probability at least $1 - e^{-\frac{1}{16}\mu_n n_1}$. Conditioned on $\mathscr{E}_2$, this probability is at least $1 - e^{-\frac{3}{128}\mu_n n}$. Since $\mu_n = 128/n$, which is admissible whenever $n \geq 256$, we conclude that $\mathscr{E}_3$ holds with probability at least $1 - e^{-3} =: 1 - \delta_3$ given $\mathscr{E}_2$.

**Wrapping up.** Taking a union bound, we have that $\mathscr{E} = \bigcup_{i=1}^3 \mathscr{E}_i$ occurs with probability at least $1 - \sum_{i=1}^3 \delta_i \geq e^{-1}/2 - e^{-12} - e^{-3} \geq 1/10$.

$\square$

## B.2 Proof of Theorem 3

### B.2.1 Overview of Results

Recall that we work in the plug-in classification setting of Section 3.1, where $\mathcal{X}$ is the feature/context space, $\mathcal{A}$ is the label/action space, and $\mathcal{D}$ is the joint distribution over context-loss pairs $(x, \ell)$. We take a class of regression functions $\mathcal{F} \subseteq (\mathcal{X} \times \mathcal{A} \to [0,1])$ as a given and make the following realizability assumption.

**Assumption 3.** *Define $f^\star(x, a) = \mathbb{E}_{\mathcal{D}}[\ell(a) \mid x]$. We assume $f^\star \in \mathcal{F}$.*

Under realizability, the optimal classifier is $\pi^\star(x) := \operatorname{argmin}_{a \in \mathcal{A}} f^\star(x, a)$, and we have $L(\pi) = \mathbb{E}[f^\star(x, \pi(x))]$. Motivated by realizability, the plug-in approach to classification finds and estimator $\widehat{f} \in \mathcal{F}$ and returns the induced classifier $\widehat{\pi}(x) := \operatorname{argmin}_{a \in \mathcal{A}} \widehat{f}(x, a)$. In this section, we estimate the losses using the following log loss regression problem.

$$\widehat{f}_{\mathsf{KL}} \leftarrow \operatorname*{argmin}_{f \in \mathcal{F}} \frac{1}{n} \sum_{i=1}^n \sum_{a \in \mathcal{A}} \ell_i(a) \log(1/f(x_i, a)) + (1 - \ell_i(a)) \log(1/(1 - f(x_i, a))). \quad (19)$$

For the resulting classifier $\widehat{\pi}_{\mathsf{KL}} := \pi_{\widehat{f}_{\mathsf{KL}}}$, we prove the following theorem.

**Theorem 3** (First-order regret bound for plug-in classification). *Let $\delta \in (0, 1)$. Suppose that Assumption 3 holds. Then with probability at least $1 - \delta$, we have*

$$L(\widehat{\pi}_{\mathsf{KL}}) - L^\star \leq 16\sqrt{\frac{L^\star \cdot A\left(\log |\mathcal{F}| + \log(A/\delta)\right)}{n}} + 68\frac{A\left(\log |\mathcal{F}| + \log(A/\delta)\right)}{n}.$$

**Multiclass classification.** We also provide a refinement of Theorem 3 for the important special case of multiclass classification. Here, rather than observing a cost function $\ell \in [0, 1]^A$ we simply observe a label $y \in \mathcal{A}$ and the goal is to predict the correct label. Formally, the distribution $\mathcal{D}$ is supported on $\mathcal{X} \times \mathcal{A}$ and we measure the error of a classifier as $\operatorname{err}(\pi) := \mathbb{P}_{\mathcal{D}}[\pi(x) \neq y]$. This can be seen as a special case of cost-sensitive classification by defining loss function $\ell(a) = \mathbb{1}\{a \neq y\}$, and the realizability assumption is as before, so that $f^\star(x, a) = \mathbb{P}_{\mathcal{D}}[a \neq y \mid x]$.

In this setting, rather than reducing to Bernoulli MLE, it is more natural to reduce to multinomial MLE. Since our function class is designed to predict the probability that a given action is *wrong* (that is, $\mathbb{P}_{\mathcal{D}}[y = a \mid x] = 1 - f^\star(x, a)$), the multinomial MLE problem is

$$\widehat{f}_{\mathsf{KL}} \leftarrow \operatorname*{argmax}_{f \in \mathcal{F}} \frac{1}{n} \sum_{i=1}^n \log(1 - f(x_i, y_i)).$$

The resulting policy is $\widehat{\pi}_{\mathsf{KL}} := \operatorname{argmin}_a \widehat{f}_{\mathsf{KL}}(x, a)$, for which we establish the following guarantee.

**Theorem 5.** *Let $\delta \in (0, 1)$ and consider the multiclass classification setting under Assumption 3. Then with probability at least $1 - \delta$,*

$$\operatorname{err}(\widehat{\pi}_{\mathsf{KL}}) - \operatorname{err}(\pi^\star) \leq 8\sqrt{\frac{\operatorname{err}(\pi^\star) \cdot 2\log(|\mathcal{F}|/\delta)}{n}} + 34\frac{\log(|\mathcal{F}|/\delta)}{n}.$$

Compared to Theorem 3, we see that by working in the simpler multiclass classification setting, we can remove the dependence on $A$ from the theorem.

### B.2.2 Preliminaries

For discrete distributions $p, q \in \Delta_A$, the Hellinger distance is defined as

$$D_{\mathrm{H}}^2(p \,\|\, q) = \frac{1}{2} \sum_a (\sqrt{p_a} - \sqrt{q_a})^2.$$

For scalars $p, q \in [0, 1]$ we overload notation and interpret $D_{\mathrm{H}}^2(p \,\|\, q) \equiv D_{\mathrm{H}}^2((p, 1-p) \,\|\, (q, 1-q))$ as the Hellinger divergence between the implied Bernoulli distributions. We similarly overload $D_{\Delta}(p \,\|\, q) \equiv D_{\Delta}((p, 1-p) \,\|\, (q, 1-q))$ as the Bernoulli triangular discrimination when given scalar arguments.

The following useful results relate Hellinger distance to the triangular discrimination for Bernoulli distributions and to a related quantity for multinomial distributions.

**Proposition 3.** *For all $p, q \in [0, 1]$, we have*

$$D_{\mathrm{H}}^2(p \,\|\, q) \geq \frac{1}{4} D_{\Delta}(p \,\|\, q) \geq \frac{1}{4} \frac{(p-q)^2}{(p+q)}.$$

**Proposition 4.** *Let $p, q \in \Delta(\mathcal{A})$ be probability mass functions. Then*

$$\max_{a \in \mathcal{A}} \frac{(p_a - q_a)^2}{(1 - p_a) + (1 - q_a)} \leq 4 D_{\mathrm{H}}^2(p \,\|\, q).$$

### B.2.3 Proof of Theorem 3 and Theorem 5

We focus on proving Theorem 3 and provide a sketch for Theorem 5, which is quite similar. For the former, the core of the argument is a generalization guarantee for $\widehat{f}_{\mathsf{KL}}$.

**Theorem 6.** *Under the conditions of Theorem 3, with probability at least $1 - \delta$, we have*

$$\mathbb{E}_{\mathcal{D}} \left[ \sum_{a \in \mathcal{A}} \frac{(\widehat{f}_{\mathsf{KL}}(x, a) - f^\star(x, a))^2}{\widehat{f}_{\mathsf{KL}}(x, a) + f^\star(x, a)} \right] \leq \frac{4A \left( \log |\mathcal{F}| + \log(A/\delta) \right)}{n}. \tag{20}$$

Theorem 6 builds on classical convergence results for maximum-likelihood estimators in well-specified settings, which provide bounds of the form

$$\mathbb{E}_{\mathcal{D}} \left[ D_{\mathrm{H}}^2(\widehat{f}_{\mathsf{KL}}(x, a) \,\|\, f^\star(x, a)) \right] \leq \mathcal{O}\left( \frac{\log(|\mathcal{F}|/\delta)}{n} \right)$$

for any fixed action [cf. 63, 71]. Theorem 6 follows quickly from this classical analysis by applying Proposition 3, which shows that the Hellinger divergence between Bernoulli distributions upper bounds the triangular discrimination that appears on the left-hand side of (20).

Theorem 3 immediately follows by combining Theorem 6 with the refined Cauchy-Schwarz lemma (Lemma 1) which we restate and prove here.

**Lemma 1** (Regret decomposition for triangular discrimination). *For any $f : \mathcal{X} \times \mathcal{A} \to [0, 1]$,*

$$L(\pi_f) - L^\star \leq 8(L^\star \cdot \mathbb{E}_{\mathcal{D}}[D_{\Delta}(f^\star(x, \cdot) \,\|\, f(x, \cdot))])^{1/2} + 17 \, \mathbb{E}_{\mathcal{D}}[D_{\Delta}(f^\star(x, \cdot) \,\|\, f(x, \cdot))]. \tag{15}$$

**Proof of Lemma 1.** Let $f \in \mathcal{F}$ be fixed. We first state a simple technical lemma.

**Lemma 3.** *For any function $f \in \mathcal{F}$ and policy $\pi : \mathcal{X} \to \mathcal{A}$,*

$$\mathbb{E}_{\mathcal{D}}[f^\star(x, \pi(x)) + f(x, \pi(x))] \leq \mathbb{E}_{\mathcal{D}}[D_{\Delta}(f^\star(x, \cdot) \,\|\, f(x, \cdot))] + 4L(\pi).$$

Going forward, define $\gamma(x, a) := f^\star(x, a) - f(x, a)$ and $s(x, a) := f^\star(x, a) + f(x, a)$, and $\Delta := \mathbb{E}_{\mathcal{D}}[D_{\Delta}(f^\star(x, \cdot) \,\|\, f(x, \cdot))]$. Let us adopt the shorthand $\mathbb{E} \equiv \mathbb{E}_{\mathcal{D}}$. We proceed to bound the cost-

sensitive regret:

$$L(\pi_f) - L(\pi^\star) \leq \mathbb{E}[f^\star(x, \pi_f(x)) - f(x, \pi_f(x)) + f(x, \pi^\star(x)) - f^\star(x, \pi^\star(x))]$$

$$\leq \mathbb{E}\left[\sqrt{\frac{\max\{s(x, \pi_f(x)), s(x, \pi^\star(x))\}}{\max\{s(x, \pi_f(x)), s(x, \pi^\star(x))\}}} \cdot (|\gamma(x, \pi_f(x))| + |\gamma(x, \pi^\star(x))|)\right]$$

$$\leq \sqrt{\mathbb{E}[\max\{s(x, \pi_f(x)), s(x, \pi^\star(x))\}]} \cdot \left(\sum_{\pi \in \{\pi_f, \pi^\star\}} \sqrt{\mathbb{E}\left[\frac{|\gamma(x, \pi(x))|^2}{\max\{s(x, \pi_f(x)), s(x, \pi^\star(x))\}}\right]}\right)$$

$$\leq \sqrt{\mathbb{E}[s(x, \pi_f(x)) + s(x, \pi^\star(x))]} \cdot \left(\sqrt{\mathbb{E}\left[\frac{\gamma(x, \pi_f(x))^2}{s(x, \pi_f(x))}\right]} + \sqrt{\mathbb{E}\left[\frac{\gamma(x, \pi^\star(x))^2}{s(x, \pi^\star(x))}\right]}\right)$$

$$\leq \sqrt{\mathbb{E}[(s(x, \pi_f(x)) + s(x, \pi^\star(x)))]} \cdot 2\sqrt{\mathbb{E}\left[\sum_a \frac{\gamma(x, a)^2}{s(x, a)}\right]}$$

$$= \sqrt{\mathbb{E}[(s(x, \pi_f(x)) + s(x, \pi^\star(x)))]} \cdot 2\sqrt{\Delta}.$$

Here, the first inequality uses that $f(x, \pi_f(x)) \leq f(x, \pi^\star(x))$ by the definition of $\pi_f$. The second inequality introduces the $s$ and $\gamma$ quantities, while the third follows from Cauchy-Schwarz. In the fourth we use that $s(x, \pi_f(x)) \leq \max\{s(x, \pi_f(x)), s(x, \pi^\star(x))\}$ and analogously for $\pi^\star$. Finally we sum over all actions to eliminate the dependence on the policies to introduce the triangular discrimination $\Delta$. Applying Lemma 3, we additionally observe that

$$\mathbb{E}\left[s(x, \pi_f(x)) + s(x, \pi^\star(x))\right] \leq 2\Delta + 4\left(L(\pi_f) + L(\pi^\star)\right).$$

After applying standard simplifications, this yields

$$L(\pi_f) - L(\pi^\star) \leq 2\sqrt{\Delta} \cdot \sqrt{2\Delta + 4(L(\pi_f) + L(\pi^\star))} \leq 2\sqrt{2}\Delta + 4\sqrt{L(\pi^\star)\Delta} + 4\sqrt{L(\pi_f)\Delta} \tag{21}$$

$$\leq 6\sqrt{2}\Delta + (L(\pi_f) + L(\pi^\star))/2.$$

Re-arranging, we deduce that $L(\pi_f) \leq 12\sqrt{2}\Delta + 3L(\pi^\star)$, and plugging this back into the first inequality in (21) gives

$$L(\pi_f) - L(\pi^\star) \leq 2\sqrt{\Delta} \cdot \sqrt{2\Delta + 4(L(\pi_f) + L(\pi^\star))} \leq 2\sqrt{\Delta} \cdot \sqrt{(2 + 48\sqrt{2})\Delta + 16L(\pi^\star)}$$

$$\leq 8\sqrt{L(\pi^\star)\Delta} + 17\Delta.$$

$\square$

**Proof sketch for Theorem 5.** The majority of the calculations in this proof are very similar to those of Theorem 3, so we highlight the two main differences. First, rather than use the triangular discrimination-type bound in Theorem 6, we use a Hellinger bound on the maximum likelihood estimate of the multinomial parameters. Specifically, using essentially the same argument as in Theorem 6, we can prove that with probability at least $1 - \delta$,

$$\mathbb{E}_{\mathcal{D}}\left[D_{\mathrm{H}}^2(\widehat{p}(\cdot \mid x) \,\|\, p^\star(\cdot \mid x))\right] \leq \frac{2\log|\mathcal{F}|/\delta}{n},$$

where $p^\star(\cdot \mid x) := \mathbb{P}_{\mathcal{D}}[y = \cdot \mid x] = 1 - f^\star(x, \cdot)$ and $\widehat{p}(\cdot \mid x) := 1 - \widehat{f}_{\mathsf{KL}}(x, \cdot)$.

The second change concerns the way we bound the quantity

$$(\widehat{f}_{\mathsf{KL}}(x, \pi(x)) - f^\star(x, \pi(x)))^2/(\widehat{f}_{\mathsf{KL}}(x, \pi(x)) + f^\star(x, \pi(x))),$$

which is done throughout the proof of Lemma 1. Rather than naively introduce a sum over all actions as was done previously, we instead apply Proposition 4, which relates the multinomial Hellinger divergence to the triangular discrimination-type quantity above. As a result, for any policy $\pi$ we have

$$\mathbb{E}_{\mathcal{D}}\left[\frac{\widehat{f}_{\mathsf{KL}}(x, \pi(x)) - f^\star(x, \pi(x)))^2}{\widehat{f}_{\mathsf{KL}}(x, \pi(x)) + f^\star(x, \pi(x))}\right] = \mathbb{E}_{\mathcal{D}}\left[\frac{(p^\star(\pi(x) \mid x) - \widehat{p}(\pi(x) \mid x))^2}{(1 - p^\star(\pi(x) \mid x)) + (1 - \widehat{p}(\pi(x) \mid x))}\right]$$

$$\leq 2\mathbb{E}_{\mathcal{D}}\left[D_{\mathrm{H}}^2(\widehat{p}(\cdot \mid x) \,\|\, p^\star(\cdot \mid x))\right].$$

All other calculations are unaffected. $\square$

### B.2.4 Proofs for Supporting Results

**Proof of Proposition 3.** Observe that we can write

$$D_{\mathrm{H}}^2(p \,\|\, q) = \frac{1}{2}(\sqrt{p} - \sqrt{q})^2 + \frac{1}{2}(\sqrt{1-p} - \sqrt{1-q})^2.$$

For each of these terms, we create a difference of squares as follows

$$(\sqrt{x} - \sqrt{y})^2 = \frac{(x-y)^2}{(\sqrt{x}+\sqrt{y})^2} \geq \frac{(x-y)^2}{2(x+y)},$$

where the last inequality uses the fact that $2\sqrt{xy} \leq x + y$. Applying this argument to both terms yields the result. $\qquad\square$

**Proof of Proposition 4.** This is an immediate consequence of the data processing inequality for Hellinger divergence and Proposition 3. Indeed, by data processing, we have

$$D_{\mathrm{H}}^2(p \,\|\, q) \geq D_{\mathrm{H}}^2((p_a, 1-p_a) \,\|\, (q_a, 1-q_a)),$$

since the latter is the distribution of the random variable $Y := \mathbb{1}\{X = a\}$ when $X \sim p$ (resp. $q$). Now that we have passed to the Bernoulli Hellinger divergence, we simply apply Proposition 3 and drop one of the two terms. $\qquad\square$

**Proof of Theorem 6.** The initial steps of this proof parallel the classical analysis of maximum likelihood estimators [see, e.g., 71]. We start by establishing a symmetrization inequality. Let $D := \{(x_i, \ell_i)\}_{i=1}^n$ and $D' := \{(x_i', \ell_i')\}_{i=1}^n$ denote two i.i.d. datasets of $n$ examples, let $C(f, D)$ be any function of a regression function $f$ and dataset $D$, and let $\widehat{f}$ be any estimator that takes the dataset $D$ and outputs a function in $\mathcal{F}$. We first show that

$$\mathbb{E}_D\left[\exp\left(C(\widehat{f}(D), D) - \log \mathbb{E}_{D'}\left[\exp(C(\widehat{f}(D), D'))\right]\right) - \log|\mathcal{F}|\right] \leq 1. \tag{22}$$

This is a symmetrization inequality because it relates the "training error" $C(\widehat{f}(D), D)$ to the error $C(\widehat{f}(D), D')$ measured on the "ghost sample" $D'$. The unusual form of the expression involving the ghost sample is to accommodate the fact that $C$ may be unbounded.

To prove (22), let $\mu$ denote the uniform distribution over $\mathcal{F}$, and observe that for any distribution $\widehat{\mu} \in \Delta(\mathcal{F})$ and any function $g : \mathcal{F} \to \mathbb{R}$, we have

$$\sum_{f \in \mathcal{F}} \widehat{\mu}(f)g(f) \leq \max_{f \in \mathcal{F}} g(f) \leq \log \sum_{f \in \mathcal{F}} \exp(g(f)) = \log\left(\mathbb{E}_{f \sim \mu} \exp(g(f))\right) + \log|\mathcal{F}|.$$

Now for any $D$ we take $\widehat{\mu}(f) := \mathbb{1}\{f = \widehat{f}(D)\}$ and $g(f) := C(f, D) - \log \mathbb{E}_{D'} \exp(C(f, D'))$ to obtain

$$C(\widehat{f}(D), D) - \log \mathbb{E}_{D'} \exp(C(\widehat{f}(D), D')) \leq \log\left(\mathbb{E}_{f \sim \mu} \frac{\exp(C(f, D))}{\mathbb{E}_{D'} \exp(C(f, D'))}\right) + \log|\mathcal{F}|.$$

We will exponentiate this inequality and take expectation over the initial dataset $D$. When we do this, the first term on the right-hand side simplifies to

$$\mathbb{E}_D \exp\left(\log\left(\mathbb{E}_{f \sim \mu}\left[\frac{\exp(C(f, D))}{\mathbb{E}_{D'} \exp(C(f, D'))}\right]\right)\right) = \mathbb{E}_{f \sim \mu}\left[\frac{\mathbb{E}_D \exp(C(f, D))}{\mathbb{E}_{D'} \exp(C(f, D'))}\right] = 1.$$

Re-arranging, we obtain (22). With the exponential moment bound in (22), a standard application of the Chernoff method yields that for any $\delta \in (0, 1)$ with probability at least $1 - \delta$ we have

$$-\log \mathbb{E}_{D'} \exp(C(\widehat{f}(D), D')) \leq -C(\widehat{f}(D), D) + \log|\mathcal{F}| + \log(1/\delta).$$

This high-probability bound holds for any fixed functional $C$. To apply it, for each $a \in \mathcal{A}$, we define

$$C_a(f, D) := -\frac{1}{2} \sum_{i=1}^n \ell_i(a) \log(f^\star(x_i, a)/f(x_i, a)) + (1 - \ell_i(a)) \log((1 - f^\star(x_i, a))/(1 - f(x_i, a))),$$

where $\ell_i(a)$ is defined as in (19). We apply the bound for each $C_a$, then take a union bound over all $a \in \mathcal{A}$ and sum up the resulting inequalities, which gives that with probability at least $1 - \delta$,

$$\sum_{a \in \mathcal{A}} - \log \mathbb{E}_{D'} \exp(C_a(\widehat{f}(D), D')) \leq \sum_{a \in \mathcal{A}} -C_a(\widehat{f}(D), D) + A \left(\log |\mathcal{F}| + \log(A/\delta)\right).$$

We will apply this inequality with $\widehat{f}_{\mathsf{KL}}$, which is the maximum likelihood estimate. Then, since $f^\star \in \mathcal{F}$ and $\widehat{f}_{\mathsf{KL}}$ minimizes the log loss, we have that $\sum_a -C_a(\widehat{f}_{\mathsf{KL}}(D), D) \leq 0$. On the other hand, for each action $a \in \mathcal{A}$, the corresponding term on the left-hand side can be simplified to

$$- \log \mathbb{E}_{D'} \exp \left( -\frac{1}{2} \sum_{i=1}^n \left( \ell_i'(a) \log \frac{f^\star(x_i', a)}{\widehat{f}_{\mathsf{KL}}(x_i', a)} + (1 - \ell_i'(a)) \log \frac{1 - f^\star(x_i', a)}{1 - \widehat{f}_{\mathsf{KL}}(x_i', a)} \right) \right)$$

Now, let $y_i'(a) \sim \mathrm{Ber}(\ell_i'(a))$. Then by Jensen's inequality, we have

$$\geq - n \log \mathbb{E}_{x', \ell'} \mathbb{E}_{y'|\ell'} \exp \left( -\frac{1}{2} \left( y'(a) \log \frac{f^\star(x', a)}{\widehat{f}_{\mathsf{KL}}(x', a)} + (1 - y'(a)) \log \frac{1 - f^\star(x', a)}{1 - \widehat{f}_{\mathsf{KL}}(x', a)} \right) \right)$$

$$= -n \log \mathbb{E}_{x', \ell'} \mathbb{E}_{y'|\ell'} \left[ \left( \frac{f^\star(x', a)}{\widehat{f}_{\mathsf{KL}}(x', a)} \right)^{-y'(a)/2} \left( \frac{1 - f^\star(x', a)}{1 - \widehat{f}_{\mathsf{KL}}(x', a)} \right)^{-(1 - y'(a))/2} \right]$$

$$= -n \log \mathbb{E}_{x'} \left[ \sqrt{f^\star(x', a) \widehat{f}_{\mathsf{KL}}(x', a)} + \sqrt{(1 - f^\star(x', a))(1 - \widehat{f}_{\mathsf{KL}}(x', a))} \right].$$

Here the last line holds because the model is well-specified; in particular $\mathbb{P}[y'(a) = 1 \mid x'] = f^\star(x', a)$. Continuing, observe that for any random variables $u, v$ taking values in $[0, 1]$ we have

$$- \log \mathbb{E} \left[ \sqrt{uv} + \sqrt{(1-u)(1-v)} \right] = - \log \left( 1 - \mathbb{E} \left[ 1 - \sqrt{uv} - \sqrt{(1-u)(1-v)} \right] \right) \geq \frac{1}{2} \mathbb{E} \left[ D_{\mathsf{H}}^2(u \| v) \right], \tag{23}$$

where the last step uses that $x \leq - \log(1 - x)$ for $x \in [0, 1]$ along with the definition of the Hellinger divergence. Together, these inequalities establish that

$$\frac{1}{2} \sum_{a \in \mathcal{A}} \mathbb{E}_x \left[ D_{\mathsf{H}}^2(f^\star(x, a) \| \widehat{f}_{\mathsf{KL}}(x, a)) \right] \leq \frac{A \left(\log |\mathcal{F}| + \log(A/\delta)\right)}{n}.$$

To conclude, we simply apply Proposition 3, which yields the result. $\qquad \square$

**Proof of Lemma 3.** Let $f \in \mathcal{F}$ be fixed and define $\gamma(x, a) := f^\star(x, a) - f(x, a)$ and $s(x, a) := f^\star(x, a) + f(x, a)$. By the triangle inequality, the AM-GM inequality, and an application of Theorem 6, we have

$$\mathbb{E}_{\mathcal{D}}[s(x, \pi(x))] \leq \mathbb{E}_{\mathcal{D}} |\gamma(x, \pi(x))| + 2L(\pi^\star)$$

$$\leq \mathbb{E}_{\mathcal{D}} \left[ \sqrt{s(x, \pi(x))} \frac{|\gamma(x, \pi(x))|}{\sqrt{s(x, \pi(x))}} \right] + 2L(\pi^\star)$$

$$\leq \frac{1}{2} \mathbb{E}_{\mathcal{D}}[s(x, \pi(x))] + \frac{1}{2} \mathbb{E}_{\mathcal{D}} \left[ \frac{\gamma(x, \pi(x))^2}{s(x, \pi(x))} \right] + 2L(\pi^\star)$$

$$\leq \frac{1}{2} \mathbb{E}_{\mathcal{D}}[s(x, \pi(x))] + \frac{1}{2} \mathbb{E}_{\mathcal{D}} \left[ \sum_a \frac{\gamma(x, a)^2}{s(x, a)} \right] + 2L(\pi^\star)$$

$$\leq \frac{1}{2} \mathbb{E}_{\mathcal{D}}[s(x, \pi(x))] + \frac{1}{2} \mathbb{E}_{\mathcal{D}}[D_\Delta(f^\star(x, \cdot) \| f(x, \cdot))] + 2L(\pi^\star).$$

Re-arranging yields the result. $\qquad \square$

# C  Proofs for Contextual Bandit Results (Section 2)

## C.1  Online Regression Oracles

In this section we briefly formalize the notion of an online regression oracle sketched in the introduction and Assumption 2. The treatment here follows Foster and Rakhlin [29].

We consider the following model for the oracle $\mathbf{Alg}_{\mathsf{KL}}$.

> For $t = 1, \dots, T$:
> - Nature selects context-action pair $(x_t, a_t) \in \mathcal{X} \times \mathcal{A}$.
> - Algorithm produces prediction $\widehat{y}_t \in [0, 1]$.
> - Nature selects outcome $y_t \in [0, 1]$.

We model the oracle as a sequence of mappings $\mathbf{Alg}_{\mathsf{KL}}^{(t)} : (\mathcal{X} \times \mathcal{A}) \times (\mathcal{X} \times \mathcal{A} \times \mathbb{R})^{t-1} \to [0, 1]$, so that $\widehat{y}_t = \mathbf{Alg}_{\mathsf{KL}}^{(t)}\big(x_t, a_t \, ; \{(x_i, , a_i, y_i)\}_{i=1}^{t-1}\big)$ above. Any algorithm of this type induces a mapping

$$\widehat{y}_t(x, a) := \mathbf{Alg}_{\mathsf{KL}}^{(t)}\big(x, a \, ; \{(x_i, , a_i, y_i)\}_{i=1}^{t-1}\big), \tag{24}$$

which may be understood as the prediction the algorithm would make at time $t$ if we froze its internal state and selected $(x_t, a_t) = (x, a)$.

## C.2  Proof of Theorem 1

**Theorem 1** (Main theorem). *Suppose Assumptions 1 and 2 hold. Then Algorithm 1 guarantees that for all sequences with $\mathbb{E}\big[\sum_{t=1}^{T} \ell_t(\pi^\star(x_t))\big] \le L^\star$, by choosing $\gamma = \sqrt{AL^\star / 3\mathbf{Reg}_{\mathsf{KL}}(T)} \vee 10A$,*

$$\mathbb{E}[\mathbf{Reg}_{\mathsf{CB}}(T)] \le 40\sqrt{L^\star \cdot A\mathbf{Reg}_{\mathsf{KL}}(T)} + 600A\mathbf{Reg}_{\mathsf{KL}}(T). \tag{6}$$

**Proof.** Define $L_T = \sum_{t=1}^{T} \ell_t(a_t)$ and $L_T^\star = \sum_{t=1}^{T} \ell_t(\pi^\star(x_t))$. All of the effort in this proof will be to show that for any choice $\gamma \ge 10A$, Algorithm 1 has

$$\mathbb{E}[\mathbf{Reg}_{\mathsf{CB}}(T)] \le \frac{10A}{\gamma} \mathbb{E}[L_T^\star] + 28\gamma \cdot \mathbf{Reg}_{\mathsf{KL}}(T). \tag{25}$$

The bound in (6) immediately follows from this guarantee by using choice of $\gamma$ in the theorem statement.

Define a filtration

$$\mathfrak{F}_{t-1} = \sigma((x_1, a_1, \ell_1(a_1)), \dots, (x_{t-1}, a_{t-1}, \ell_{t-1}(a_{t-1})), x_t) \tag{26}$$

and let $\mathbb{E}_t[\cdot] := \mathbb{E}[\cdot \mid \mathfrak{F}_t]$. Next, define the following conditional-expected versions of the contextual bandit regret and log loss regret, respectively

$$\overline{\mathbf{Reg}}_{\mathsf{CB}}(T) = \sum_{t=1}^{T} \mathbb{E}_{t-1}[\ell_t(a_t) - \ell_t(\pi^\star(x_t))] = \sum_{t=1}^{T} \sum_{a} p_{t,a}(f^\star(x_t, a) - f^\star(x_t, \pi^\star(x_t)))$$

and

$$\overline{\mathbf{Reg}}_{\mathsf{KL}}(T) = \sum_{t=1}^{T} \mathbb{E}_{t-1}[\ell_{\log}(\widehat{y}_t(x_t, a_t), \ell_t(a_t)) - \ell_{\log}(f^\star(x_t, a_t), \ell_t(a_t))].$$

Our starting point is to observe that $\mathbb{E}[\mathbf{Reg}_{\mathsf{CB}}(T)] = \mathbb{E}\big[\overline{\mathbf{Reg}}_{\mathsf{CB}}(T)\big]$ and $\mathbb{E}\big[\overline{\mathbf{Reg}}_{\mathsf{KL}}(T)\big] \le \mathbf{Reg}_{\mathsf{KL}}(T)$, where the latter holds since $\mathbf{Reg}_{\mathsf{KL}}(T)$ is a deterministic upper bound on the log loss regret of the oracle. So it suffices to relate the conditional-expected versions of these quantities.

The main step of the proof is to upper bound $\overline{\mathbf{Reg}}_{\mathsf{CB}}(T)$, using the first-order per-round inequality Theorem 4 (proven in Appendix C.3), which we restate here for completeness.

**Theorem 4** (First-order per-round inequality). *Let $y \in [0, 1]^A$ be given and $b \in \operatorname{argmin}_a y_a$. Define $p_a = \frac{y_b}{Ay_b + \gamma(y_a - y_b)}$ for $a \neq b$, and $p_b = 1 - \sum_{a \neq b} p_a$. If $\gamma \geq 2A$, then for all $f \in [0, 1]^A$ and $a^\star \in \operatorname{argmin}_a f_a$, we have*

$$\underbrace{\sum_a p_a(f_a - f_{a^\star})}_{\text{CB regret}} \leq \underbrace{\frac{5A}{\gamma} \sum_a p_a f_a}_{\text{bias from }\textit{exploring}} + \underbrace{7\gamma \sum_a p_a \frac{(y_a - f_a)^2}{y_a + f_a}}_{\text{error from }\textit{exploiting}} . \tag{17}$$

Applying Theorem 4 for each round $t$, we are guaranteed that

$$\overline{\mathbf{Reg}}_{\mathsf{CB}}(T) \leq \frac{5A}{\gamma} \sum_{t=1}^T \sum_a p_{t,a} f^\star(x_t, a) + 7\gamma \sum_{t=1}^T \sum_a p_{t,a} \frac{(\widehat{y}_t(x_t, a) - f^\star(x_t, a))^2}{\widehat{y}_t(x_t, a) + f^\star(x_t, a)}$$

$$= \frac{5A}{\gamma} \overline{L}_T + 7\gamma \cdot \overline{\mathbf{Err}}_\Delta(T),$$

where $\overline{L}_T := \sum_{t=1}^T \sum_a p_{t,a} f^\star(x_t, a)$ and

$$\overline{\mathbf{Err}}_\Delta(T) := \sum_{t=1}^T \sum_a p_{t,a} \frac{(\widehat{y}_t(x_t, a) - f^\star(x_t, a))^2}{\widehat{y}_t(x_t, a) + f^\star(x_t, a)}.$$

Next, we relate the triangular discrimination-type error $\overline{\mathbf{Err}}_\Delta(T)$ to the log loss regret using the following proposition (proven in the sequel).

**Proposition 5.** *If $y \in [0, 1]$ is a random variable with $\mathbb{E}[y] = \mu$, then for any $\widehat{y} \in [0, 1]$,*

$$\mathbb{E}[\ell_{\log}(\widehat{y}, y) - \ell_{\log}(\mu, y)] = d_{\mathrm{KL}}(\mu \,\|\, \widehat{y}) \geq \frac{1}{2} \cdot \frac{(\widehat{y} - \mu)^2}{\widehat{y} + \mu}. \tag{27}$$

In particular, since $a_t$ and $\ell_t$ are conditionally independent given $\mathfrak{F}_{t-1}$, this implies that

$$\overline{\mathbf{Err}}_\Delta(T) \leq 2 \sum_{t=1}^T \sum_a p_{t,a} d_{\mathrm{KL}}(f^\star(x_t, a) \,\|\, \widehat{y}_t(x, a_t)) = 2\overline{\mathbf{Reg}}_{\mathsf{KL}}(T),$$

so that

$$\overline{\mathbf{Reg}}_{\mathsf{CB}}(T) \leq \frac{5A}{\gamma} \overline{L}_T + 14\gamma \cdot \overline{\mathbf{Reg}}_{\mathsf{KL}}(T).$$

To conclude, let $\overline{L}_T^\star = \sum_{t=1}^T f^\star(x_t, \pi^\star(x_t))$. Then this inequality can be written as

$$\overline{L}_T - \overline{L}_T^\star \leq \frac{5A}{\gamma} \overline{L}_T + 14\gamma \cdot \overline{\mathbf{Reg}}_{\mathsf{KL}}(T).$$

Since $1/(1 - \varepsilon) \leq 1 + 2\varepsilon$ for all $\varepsilon \leq 1/2$, this implies that whenever $\gamma \geq 10A$,

$$\overline{L}_T - \overline{L}_T^\star \leq \frac{10A}{\gamma} \overline{L}_T^\star + 28\gamma \cdot \overline{\mathbf{Reg}}_{\mathsf{KL}}(T).$$

Noting that $\mathbb{E}[\overline{L}_T^\star] = \mathbb{E}[L_T^\star]$ and $\mathbb{E}[\overline{L}_T] = \mathbb{E}[L_T]$, this establishes (25).

$\square$

**Proof of Proposition 5.** For the equality in (27), we have

$$\mathbb{E}[\ell_{\log}(\widehat{y}, y) - \ell_{\log}(\mu, y)] = \mathbb{E}[y \log(\mu/\widehat{y}) + (1 - y) \log((1 - \mu)/(1 - \widehat{y}))] = d_{\mathrm{KL}}(\mu \,\|\, \widehat{y}).$$

To prove the inequality, let $f_{\widehat{y}}(\mu) = d_{\mathrm{KL}}(\mu \,\|\, \widehat{y})$. By Taylor's theorem, we have

$$f_{\widehat{y}}(\mu) = f_{\widehat{y}}(\widehat{y}) + f'_{\widehat{y}}(\widehat{y})(\mu - \widehat{y}) + \frac{1}{2} f''_{\widehat{y}}(\bar{y})(\mu - \widehat{y})^2,$$

for some $\bar{y} \in \operatorname{conv}(\{\widehat{y}, \mu\})$. Observe that

$$f'_{\widehat{y}}(z) = \log(z/\widehat{y}) - \log((1 - z)/(1 - \widehat{y})),$$

so that we have $f_{\widehat{y}}(\widehat{y}) = f'_{\widehat{y}}(\widehat{y}) = 0$. Further

$$f''_{\widehat{y}}(\bar{y}) = \frac{1}{\bar{y}} + \frac{1}{1 - \bar{y}} \geq \frac{1}{\max\{\widehat{y}, \mu\}} \geq \frac{1}{\widehat{y} + \mu},$$

which establishes the result. $\square$

### C.3 Proof of Theorem 4

**Theorem 4** (First-order per-round inequality). *Let $y \in [0, 1]^A$ be given and $b \in \arg\min_a y_a$. Define $p_a = \frac{y_b}{Ay_b + \gamma(y_a - y_b)}$ for $a \neq b$, and $p_b = 1 - \sum_{a \neq b} p_a$. If $\gamma \geq 2A$, then for all $f \in [0, 1]^A$ and $a^\star \in \arg\min_a f_a$, we have*

$$\underbrace{\sum_a p_a(f_a - f_{a^\star})}_{\text{CB regret}} \leq \underbrace{\frac{5A}{\gamma} \sum_a p_a f_a}_{\text{bias from } exploring} + \underbrace{7\gamma \sum_a p_a \frac{(y_a - f_a)^2}{y_a + f_a}}_{\text{error from } exploiting}. \tag{17}$$

**Proof.** To begin, we observe that by the AM-GM inequality,

$$\sum_a p_a(f_a - f_{a^\star}) = \sum_{a \neq a^\star} p_a(y_a - f_{a^\star}) + \sum_{a \neq a^\star} p_a(f_a - y_a)$$

$$\leq \sum_{a \neq a^\star} p_a(y_a - f_{a^\star}) + \frac{1}{4\gamma} \sum_{a \neq a^\star} p_a(f_a + y_a) + \gamma \sum_{a \neq a^\star} p_a \frac{(y_a - f_a)^2}{y_a + f_a}. \tag{28}$$

We focus on bounding the first term in (28), then return to the other terms at the end of the proof. We have

$$\sum_{a \neq a^\star} p_a(y_a - f_{a^\star}) = \sum_{a \neq a^\star} p_a(y_a - y_b) + (1 - p_{a^\star})(y_b - f_{a^\star})$$

$$= \sum_{a \notin \{a^\star, b\}} p_a(y_a - y_b) + (1 - p_{a^\star})(y_b - f_{a^\star}). \tag{29}$$

Recall that for $a \neq b$ we set $p_a = \frac{y_b}{Ay_b + \gamma(y_a - y_b)}$ and for $p_b$ we set $p_b = 1 - \sum_{a \neq b} p_a$. With this setting, the first term in (29) is bounded as

$$\sum_{a \notin \{a^\star, b\}} p_a(y_a - y_b) \leq \sum_{a \notin \{a^\star, b\}} \frac{y_b(y_a - y_b)}{Ay_b + \gamma(y_a - y_b)} \leq A\frac{y_b}{\gamma}. \tag{30}$$

It remains to bound the term

$$(1 - p_{a^\star})(y_b - f_{a^\star}).$$

If $f_{a^\star} \geq y_b$ this is trivially negative, so we assume going forward that $f_{a^\star} \leq y_b$, and upper bound as

$$(1 - p_{a^\star})(y_b - f_{a^\star}) \leq y_b - f_{a^\star}.$$

We now appeal to the following lemma.

**Lemma 4.** *The distribution $p$ in Theorem 4 ensures that*

$$y_b - f_{a^\star} \leq \frac{A}{4\gamma} y_b + 2\gamma \cdot p_{a^\star} \frac{(y_{a^\star} - f_{a^\star})^2}{y_{a^\star} + f_{a^\star}}. \tag{31}$$

Combining (28), (30), and (31), we arrive at the bound.

$$\sum_a p_a(f_a - f_{a^\star}) \leq \frac{1}{4\gamma} \sum_a p_a(f_a + y_a) + 2\gamma \sum_a p_a \frac{(y_a - f_a)^2}{y_a + f_a} + \frac{2A}{\gamma} y_b. \tag{32}$$

To conclude, we relate the non-triangular terms above to $\sum_a p_a f_a$, which corresponds to the learner's expected loss. For the first term, we use the following basic result.

**Lemma 5.** *For any distribution $p \in \Delta_A$,*

$$\sum_a p_a y_a \leq 3 \sum_a p_a f_a + \sum_a p_a \frac{(y_a - f_a)^2}{y_a + f_a}.$$

Applying this gives

$$\sum_a p_a(f_a - f_{a^\star}) \leq \frac{1}{\gamma} \sum_a p_a f_a + 3\gamma \sum_a p_a \frac{(y_a - f_a)^2}{y_a + f_a} + \frac{2A}{\gamma} y_b,$$

where we have used that $\gamma \geq 1$ to simplify. Our final step is to relate the last term above to $f_{a^\star}$. To do this, we observe that if $\gamma \geq 2A$, then Lemma 4 implies (after rearranging), that

$$y_b \leq 2f_{a^\star} + 4\gamma \cdot p_{a^\star} \frac{(y_{a^\star} - f_{a^\star})^2}{y_{a^\star} + f_{a^\star}},$$

so that

$$\frac{2A}{\gamma} y_b \leq \frac{4A}{\gamma} f_{a^\star} + 8A p_{a^\star} \frac{(y_{a^\star} - f_{a^\star})^2}{y_{a^\star} + f_{a^\star}} \leq \frac{4A}{\gamma} f_{a^\star} + 4\gamma \cdot p_{a^\star} \frac{(y_{a^\star} - f_{a^\star})^2}{y_{a^\star} + f_{a^\star}}.$$

With this, we have

$$\sum_a p_a (f_a - f_{a^\star}) \leq \frac{1}{\gamma} \sum_a p_a f_a + 7\gamma \sum_a p_a \frac{(y_a - f_a)^2}{y_a + f_a} + \frac{4A}{\gamma} f_{a^\star},$$

Finally, since $a^\star \in \operatorname{argmin}_a f_a$, we have $f_{a^\star} \leq \sum_a p_a f_a$, so we can simplify to

$$\sum_a p_a (f_a - f_{a^\star}) \leq \frac{5A}{\gamma} \sum_a p_a f_a + 7\gamma \sum_a p_a \frac{(y_a - f_a)^2}{y_a + f_a}.$$

$\square$

### C.3.1 Proofs for Supporting Lemmas

**Proof of Lemma 4.** Assume that $y_b \geq f_{a^\star}$, or else we are done. We consider two cases.

**Case 1:** $a^\star = b$. In this case, by the AM-GM inequality

$$y_b - f_{a^\star} = y_{a^\star} - f_{a^\star} \leq \frac{y_{a^\star} + f_{a^\star}}{8\gamma p_{a^\star}} + 2\gamma \cdot p_{a^\star} \frac{(y_{a^\star} - f_{a^\star})^2}{y_{a^\star} + f_{a^\star}}.$$

Since $a^\star = b$, we have

$$p_{a^\star} = p_b = 1 - \sum_{a \neq b} \frac{y_b}{Ay_b + \gamma(y_a - y_b)} \geq 1/A,$$

so we can further upper bound by

$$\frac{A}{8\gamma}(y_{a^\star} + f_{a^\star}) + 2\gamma \cdot p_{a^\star} \frac{(y_{a^\star} - f_{a^\star})^2}{y_{a^\star} + f_{a^\star}} \leq \frac{A}{4\gamma} y_{a^\star} + 2\gamma \cdot p_{a^\star} \frac{(y_{a^\star} - f_{a^\star})^2}{y_{a^\star} + f_{a^\star}} = \frac{A}{4\gamma} y_b + 2\gamma \cdot p_{a^\star} \frac{(y_{a^\star} - f_{a^\star})^2}{y_{a^\star} + f_{a^\star}},$$

where we have used that $f_{a^\star} \leq y_b = y_{a^\star}$, where the latter holds since, for this case, we are assuming $a^\star = b$.

**Case 2:** $a^\star \neq b$. Observe that in this case, we have

$$y_{a^\star} \geq y_b, \quad \text{and} \quad f_b \geq f_{a^\star}. \tag{33}$$

Since $a^\star \neq b$, using the definition of $p_{a^\star}$, we have

$$y_b - f_{a^\star} = p_{a^\star} \frac{Ay_b + \gamma(y_{a^\star} - y_b)}{y_b}(y_b - f_{a^\star})$$

$$= A p_{a^\star}(y_b - f_{a^\star}) + \gamma \cdot p_{a^\star} \frac{(y_{a^\star} - y_b)(y_b - f_{a^\star})}{y_b},$$

which we can rewrite as

$$y_b - f_{a^\star} = \underbrace{A p_{a^\star}(y_b - f_{a^\star}) - \gamma \cdot p_{a^\star} \frac{(y_b - f_{a^\star})^2}{y_b}}_{\mathbf{A}} + \underbrace{\gamma \cdot p_{a^\star} \frac{(y_{a^\star} - f_{a^\star})(y_b - f_{a^\star})}{y_b}}_{\mathbf{B}}.$$

For the term **A** above, we observe that by the AM-GM inequality,

$$A p_{a^\star}(y_b - f_{a^\star}) \leq \frac{A^2}{4\gamma} p_{a^\star} y_b + \gamma p_{a^\star} \frac{(y_b - f_{a^\star})^2}{y_b}, \tag{34}$$

so that

$$\mathbf{A} \le \frac{A^2}{4\gamma} p_{a^\star} y_b \le \frac{A}{4\gamma} y_b,$$

where we have used that $p_{a^\star} \le 1/A$ when $a^\star \neq b$.

Next, to bound $\mathbf{B}$, we observe that $y_{a^\star} \ge y_b \ge f_{a^\star} \ge 0$. Since the function $a \mapsto \frac{(a-b)}{a}$ is increasing for $a, b \ge 0$, we have that $\frac{(y_b - f_{a^\star})}{y_b} \le \frac{(y_{a^\star} - f_{a^\star})}{y_{a^\star}}$ and consequently

$$\frac{(y_{a^\star} - f_{a^\star})(y_b - f_{a^\star})}{y_b} \le \frac{(y_{a^\star} - f_{a^\star})^2}{y_{a^\star}} \le 2\frac{(y_{a^\star} - f_{a^\star})^2}{y_{a^\star} + f_{a^\star}},$$

where the second inequality uses that $y_{a^\star} \ge f_{a^\star}$.

Altogether, we have that when $a^\star \neq b$,

$$y_b - f_{a^\star} = \mathbf{A} + \mathbf{B} \le \frac{A}{4\gamma} y_b + 2\gamma \cdot p_{a^\star} \frac{(y_{a^\star} - f_{a^\star})^2}{y_{a^\star} + f_{a^\star}}. \tag{35}$$

The result now follows by combining the two cases. $\qquad\square$

**Proof of Lemma 5.** First, we write

$$\sum_a p_a y_a = \sum_a p_a f_a + \sum_a p_a (y_a - f_a).$$

By the AM-GM inequality, we have

$$\sum_a p_a (y_a - f_a) \le \frac{1}{2} \sum_a p_a (y_a + f_a) + \frac{1}{2} \sum_a p_a \frac{(y_a - f_a)^2}{y_a + f_a},$$

so that

$$\sum_a p_a y_a \le \frac{1}{2} \sum_a p_a y_a + \frac{3}{2} \sum_a p_a f_a + \frac{1}{2} \sum_a p_a \frac{(y_a - f_a)^2}{y_a + f_a},$$

and after rearranging,

$$\sum_a p_a y_a \le 3 \sum_a p_a f_a + \sum_a p_a \frac{(y_a - f_a)^2}{y_a + f_a}.$$

$\qquad\square$

## C.4 Auxiliary Results

**Proposition 6.** *When $\widehat{y}, y \in [0,1]$, the logarithmic loss $\widehat{y} \mapsto \ell_{\log}(\widehat{y}, y)$ is 1-exp-concave and 1-mixable.*

**Proof of Proposition 6.** Let $f_y(\widehat{y}) = \ell_{\log}(\widehat{y}, y)$. From Hazan et al. [38], the loss is $\alpha$-exp-concave if and only if $f_y''(\widehat{y}) \ge \alpha(f_y'(\widehat{y}))^2$ for all $\widehat{y}, y \in [0,1]$. We observe that $f_y'(\widehat{y}) = -\frac{y}{\widehat{y}} + \frac{1-y}{1-\widehat{y}}$ and $f_y''(\widehat{y}) = \frac{y}{\widehat{y}^2} + \frac{1-y}{(1-\widehat{y})^2}$. Since $y \in [0,1]$, Jensen's inequality implies that

$$(f_y'(\widehat{y}))^2 \le y\left(\frac{-1}{\widehat{y}}\right)^2 + (1-y)\left(\frac{1}{1-\widehat{y}}\right)^2 = f_y''(\widehat{y}),$$

so we may take $\alpha = 1$.

Mixability is an immediate consequence of exp-concavity [21]. $\qquad\square$

# D  Extensions

## D.1  Small Rewards

In this section we sketch an extension of FastCB to the setting where the learner observes rewards $r_t(a) \in [0, 1]$ rather than losses $\ell_t(a)$, and aims to achieve high reward rather than low loss. As before, we assume access to a function class $\mathcal{F}$ such that the Bayes predictor $f^\star(x, a) := \mathbb{E}[r(a) \mid x] \in \mathcal{F}$. Formally, we define regret for this setting as

$$\mathbf{Reg}_{\mathsf{CB}}(T) = \sum_{t=1}^{T} r_t(\pi^\star(x_t)) - \sum_{t=1}^{T} r_t(a_t),$$

where $\pi^\star(x) := \mathrm{argmax}_{a \in \mathcal{A}} f^\star(x, a)$ is the optimal policy.

Our aim here is to provide regret bounds that adapt whenever the reward of the optimal policy is small. This type of guarantee is natural if we believe a-priori that rewards are typically very small, which is common in personalization and recommendation applications, where clicks are often used as reward signal, yet click-through rates are typically well below $1\%$. In such settings, it is favorable to have regret scaling with the reward $R^\star$ of the optimal policy. Note that this is *not* equivalent to an $L^\star$ bound after the translation $r_t(a) = 1 - \ell_t(a)$, since having low reward corresponds to having high loss.

FastCB can be adapted to the small-reward setting achieve

$$\mathbb{E}[\mathbf{Reg}_{\mathsf{CB}}(T)] \leq \mathcal{O}\left(\sqrt{R^\star \cdot A\mathbf{Reg}_{\mathsf{KL}}(T)} + A\mathbf{Reg}_{\mathsf{KL}}(T)\right)$$

whenever $\mathbb{E}\left[\sum_{t=1}^{T} r_t(\pi^\star(x_t))\right] \leq R^\star$. The algorithm remains essentially as described in Algorithm 1, with the only difference being that we change the reweighted inverse gap weighting strategy used in Line 6. The new strategy and corresponding per-round inequality are described in the following theorem.

**Theorem 7.** *Let $y \in [0, 1]^A$ be given and $b := \mathrm{argmax}_a \, y_a$. Define $p_a = \frac{y_b}{Ay_b + \gamma(y_b - y_a)}$ for $a \neq b$ and $p_b = 1 - \sum_{a \neq b} p_a$. If $\gamma \geq 4A$, then for all $f \in [0, 1]^A$ and $a^\star \in \mathrm{argmax}_a \, f_a$, we have*

$$\sum_a p_a(f_{a^\star} - f_a) \leq \frac{9A}{\gamma} \sum_a p_a f_a + 10\gamma \sum_a p_a \frac{(y_a - f_a)^2}{y_a + f_a}.$$

Observe that the left hand side is the per-round regret of the learner when $f$ is the *reward* (rather than loss) model, which contrasts with the left-hand side in Theorem 4. On the other hand, the right-hand side only differs from that of Theorem 4 in the constants. As such, it naturally yields an $R^\star$ bound when applied with $y = \widehat{y}_t(x_t, \cdot)$ as in Algorithm 1.

It should be noted that achieving $R^\star$-based first-order bounds for contextual bandits appears to be considerably easier than achieving $L^\star$-based bounds. Indeed, the standard analysis of the Exp4 algorithm already yields a $\mathcal{O}(\sqrt{R^\star \cdot A \log |\Pi|})$ regret bound, under the benign assumption that the policy class contains the policy that selects actions uniformly at random on every context [9, Theorem 7.1]. On the other hand, Exp4 cannot achieve an $L^\star$-based bound without modifications [6].

**Proof of Theorem 7.**  The proof parallels that of Theorem 4. We start by adding and subtracting $y_a$ and applying the AM-GM inequality

$$\sum_a p_a(f_{a^\star} - f_a) = \sum_{a \neq a^\star} p_a(f_{a^\star} - y_a) + \sum_{a \neq a^\star} p_a(y_a - f_a)$$

$$\leq \sum_{a \neq a^\star} p_a(f_{a^\star} - y_a) + \frac{1}{4\gamma} \sum_{a \neq a^\star} p_a(y_a + f_a) + \gamma \sum_{a \neq a^\star} p_a \frac{(y_a - f_a)^2}{y_a + f_a}.$$

For the first term above, let us consider two cases.

**Case 1.**  First, if $y_b \geq f_{a^\star}$ then

$$\sum_{a \neq a^\star} p_a(f_{a^\star} - y_a) \leq \sum_{a \notin \{a^\star, b\}} p_a(f_{a^\star} - y_a) \leq \sum_{a \notin \{a^\star, b\}} p_a(f_{a^\star} - y_a)\mathbb{1}\{f_{a^\star} \geq y_a\}.$$

Here we have simply dropped negative terms. Now, using the definition of $p_a$ for $a \neq b$, we have

$$p_a(f_{a^\star} - y_a) \mathbb{1}\{f_{a^\star} \geq y_a\} = \frac{y_b(f_{a^\star} - y_a)}{Ay_b + \gamma(y_b - y_a)} \mathbb{1}\{f_{a^\star} \geq y_a\} \leq \frac{y_b(f_{a^\star} - y_a)}{\gamma(y_b - y_a)} \mathbb{1}\{f_{a^\star} \geq y_a\}.$$

Observe that $y_b/(y_b - y_a) \leq f_{a^\star}/(f_{a^\star} - y_a)$, since $y_b \geq f_{a^\star} \geq y_a \geq 0$. This yields

$$\mathbb{1}\{f_{a^\star} \geq y_a\} \frac{y_b(f_{a^\star} - y_a)}{\gamma(y_b - y_a)} \leq \mathbb{1}\{f_{a^\star} \geq y_a\} \frac{f_{a^\star}(f_{a^\star} - y_a)}{\gamma(f_{a^\star} - y_a)} \leq \frac{f_{a^\star}}{\gamma}.$$

And so, if $y_b \geq f_{a^\star}$ we have the bound

$$\sum_a p_a(f_{a^\star} - f_a) \leq \frac{Af_{a^\star}}{\gamma} + \frac{1}{4\gamma} \sum_{a \neq a^\star} p_a(f_a + y_a) + \gamma \sum_{a \neq a^\star} p_a \frac{(y_a - f_a)^2}{y_a + f_a}.$$

**Case 2.** If $y_b \leq f_{a^\star}$ then for the first term, we write

$$\sum_{a \neq a^\star} p_a(f_{a^\star} - y_a) = \sum_{a \notin \{a^\star, b\}} p_a(y_b - y_a) + (1 - p_{a^\star})(f_{a^\star} - y_b) \leq \sum_{a \notin \{a^\star, b\}} p_a(y_b - y_a) + (f_{a^\star} - y_b).$$

$$(36)$$

For the first term in (36), using the definition of $p_a$, we have

$$\sum_{a \notin \{a^\star, b\}} p_a(y_b - y_a) = \sum_{a \notin \{a^\star, b\}} \frac{y_b(y_b - y_a)}{Ay_b + \gamma(y_b - y_a)} \leq \sum_{a \notin \{a^\star, b\}} \frac{y_b}{\gamma} \leq \frac{Ay_b}{\gamma} \leq \frac{Af_{a^\star}}{\gamma}. \qquad (37)$$

For the second term, we first note that $p_b = 1 - \sum_{a \neq b} p_a \geq 1 - \sum_{a \neq b} \frac{y_b}{Ay_b} \geq \frac{1}{A}$, then consider two subcases.

**Case 2a ($y_b \leq f_{a^\star}$ and $a^\star = b$).** Here we simply use the AM-GM inequality to show that

$$f_{a^\star} - y_b = f_{a^\star} - y_{a^\star} \leq \frac{f_{a^\star} + y_{a^\star}}{8\gamma p_{a^\star}} + 2\gamma p_{a^\star} \frac{(y_{a^\star} - f_{a^\star})^2}{y_{a^\star} + f_{a^\star}}$$

$$\leq \frac{A}{4\gamma} f_{a^\star} + 2\gamma p_{a^\star} \frac{(y_{a^\star} - f_{a^\star})^2}{y_{a^\star} + f_{a^\star}}.$$

Here the first inequality is AM-GM, while the second uses that $y_{a^\star} = y_b \leq f_{a^\star}$ (by the conditions for this case), along with the fact that $p_{a^\star} = p_b \geq 1/A$.

**Case 2b ($y_b \leq f_{a^\star}$ and $a^\star \neq b$).** In this case, we have

$$y_b \geq y_{a^\star}, \quad \text{and} \quad f_{a^\star} \geq f_b.$$

Using the definition for $p_{a^\star}$, we have

$$f_{a^\star} - y_b = p_{a^\star} \frac{Ay_b + \gamma(y_b - y_{a^\star})}{y_b}(f_{a^\star} - y_b) = p_{a^\star}A(f_{a^\star} - y_b) + p_{a^\star}\gamma \frac{(y_b - y_{a^\star})(f_{a^\star} - y_b)}{y_b}$$

$$\leq p_{a^\star}A(f_{a^\star} - y_b) + p_{a^\star}\gamma \frac{(f_{a^\star} - y_{a^\star})(f_{a^\star} - y_b)}{f_{a^\star}}$$

$$= p_{a^\star}A(f_{a^\star} - y_b) + p_{a^\star}\gamma \frac{(f_{a^\star} - y_{a^\star})^2}{f_{a^\star}} + p_{a^\star}\gamma \frac{(f_{a^\star} - y_{a^\star})(y_{a^\star} - y_b)}{f_{a^\star}}$$

$$\leq p_{a^\star}A(f_{a^\star} - y_b) + p_{a^\star}\gamma \frac{(f_{a^\star} - y_{a^\star})^2}{f_{a^\star}}$$

$$\leq p_{a^\star}A(f_{a^\star} - y_{a^\star}) + p_{a^\star}\gamma \frac{(f_{a^\star} - y_{a^\star})^2}{f_{a^\star}}.$$

Here, in the first inequality we use that $a \mapsto (a - b)/a$ is increasing in $a$, for $a, b \geq 0$ along with the fact that $f_{a^\star} \geq y_b \geq y_{a^\star}$. The second and third inequalities both use that $y_{a^\star} \leq y_b$.

Now by the AM-GM inequality, we have

$$p_{a^\star} A(f_{a^\star} - y_{a^\star}) \leq \frac{p_{a^\star} A^2}{4\gamma} f_{a^\star} + p_{a^\star} \gamma \frac{(f_{a^\star} - y_{a^\star})^2}{f_{a^\star}}$$

$$\leq \frac{A}{4\gamma} f_{a^\star} + p_{a^\star} \gamma \frac{(f_{a^\star} - y_{a^\star})^2}{f_{a^\star}},$$

where the second inequality uses the fact that $p_{a^\star} \leq 1/A$ since $a_\star \neq b$. Finally, we use that $y_{a^\star} \leq f_{a^\star}$ to conclude that in this case,

$$f_{a^\star} - y_b \leq \frac{A}{4\gamma} f_{a^\star} + 4\gamma p_{a^\star} \frac{(f_{a^\star} - y_{a^\star})^2}{f_{a^\star} + y_{a^\star}}. \tag{38}$$

This bound applies to both Case 2a and 2b.

**Wrapping up.**    Returning to Case 2 and combining (36), (37), and (38), we have

$$\sum_{a \neq a^\star} p_a(f_{a^\star} - y_a) \leq \frac{2A f_{a^\star}}{\gamma} + 4\gamma p_{a^\star} \frac{(f_{a^\star} - y_{a^\star})^2}{f_{a^\star} + y_{a^\star}}.$$

Combining this with our initial calculation, we have

$$\sum_a p_a(f_{a^\star} - f_a) \leq \frac{2A f_{a^\star}}{\gamma} + 4\gamma p_{a^\star} \frac{(f_{a^\star} - y_{a^\star})^2}{f_{a^\star} + y_{a^\star}} + \frac{1}{4\gamma} \sum_{a \neq a^\star} p_a(y_a + f_a) + \gamma \sum_{a \neq a^\star} p_a \frac{(y_a - f_a)^2}{y_a + f_a}$$

$$\leq 4\gamma \sum_a p_a \frac{(y_a - f_a)^2}{y_a + f_a} + \frac{1}{4\gamma} \sum_a p_a(y_a + f_a) + \frac{2A}{\gamma} f_{a^\star}.$$

Next, we can apply Lemma 5 as-is, which yields

$$\sum_a p_a(f_{a^\star} - f_a) \leq \frac{1}{\gamma} \sum_a p_a f_a + 5\gamma \sum_a p_a \frac{(y_a - f_a)^2}{y_a + f_a} + \frac{2A}{\gamma} f_{a^\star}.$$

This inequality, after using assumption the that $\gamma \geq 4A$ and rearranging, implies

$$f_{a^\star} \leq 2(1 + 1/\gamma) \sum_a p_a f_a + 10\gamma \sum_a p_a \frac{(y_a - f_a)^2}{y_a + f_a} \leq 4 \sum_a p_a f_a + 10\gamma \sum_a p_a \frac{(y_a - f_a)^2}{y_a + f_a}.$$

Plugging this bound in for the final expression gives

$$\sum_a p_a(f_{a^\star} - f_a) \leq \frac{1}{\gamma} \sum_a p_a f_a + 5\gamma \sum_a p_a \frac{(y_a - f_a)^2}{y_a + f_a} + \frac{8A}{\gamma} \sum_a p_a f_A + 20A \sum_a p_a \frac{(y_a - f_a)^2}{y_a + f_a}$$

$$\leq \frac{9A}{\gamma} \sum_a p_a f_a + 10\gamma \sum_a p_a \frac{(y_a - f_a)^2}{y_a + f_a},$$

as desired. □

# E    Details for Experiments

## E.1    Assets and Computing Resources

**Assets.**    The code for the contextual bandit evaluation setup of Bietti et al. [13], which we used as a starting point, is publicly available at https://github.com/albietz/cb_bakeoff. Likewise, the source code for Vowpal Wabbit, upon which our implementation is built, is publicly available at https://github.com/vowpalwabbit/vowpal_wabbit/. The source code used to run the experiments is included in the supplementary material.

All datasets used in the experiments are publicly available via the OpenML collection (https://www.openml.org). Readers can refer to the information page for each respective dataset (e.g., https://www.openml.org/d/1041 for dataset 1041) for copyright information.

**Computing resources.** Experiments were run on a single `n1-highcpu-32` instance on Google Compute Engine. The total compute time required to run the experiments was under 12 hours.

## E.2 Additional Details

**Datasets.** We restrict to a subset of the bake-off suite consisting of 516 multiclass classification datasets in the same fashion as Foster et al. [35].

**Algorithms and oracle.** For SquareCB.L and FastCB.L we take $\mathcal{F}$ to be a class of generalized linear models:

$$\mathcal{F} = \big\{ (x, a) \mapsto \sigma(\langle w, \phi(x, a) \rangle) \mid w \in \mathbb{R}^d \big\}, \tag{39}$$

where $\sigma(t) = 1/(1 + e^{-t})$ is the logistic link function and $\phi(x, a)$ is a fixed (dataset-dependent) feature map. This choice is convenient because i) it naturally produces predictions in $[0, 1]$, as required by FastCB, and ii), we have that $\ell_{\log}(\sigma(\langle w, \phi(x, a) \rangle), y) = \ell_{\text{logistic}}(\langle w, \phi(x, a) \rangle, y)$, so that online regression with the logarithmic loss is equivalent to online logistic regression (cf. Example 4).

Even though SquareCB is designed for the square loss rather than the log loss, one can show that under the realizability assumption (Assumption 1), any log loss oracle is admissible for SquareCB. Indeed, for any log loss oracle satisfying Assumption 2, realizability and Pinsker's inequality imply that

$$\mathbb{E}\left[\sum_{t=1}^{T}(\widehat{y}_t(x_t, a_t) - f^\star(x_t, a_t))^2\right] \leq 2\,\mathbb{E}\left[\sum_{t=1}^{T} d_{\text{KL}}(f^\star(x_t, a_t) \,\|\, \widehat{y}_t(x_t, a_t))\right] \leq 2\mathbf{Reg}_{\text{KL}}(T), \tag{40}$$

which means that the oracle is a valid square loss oracle for SquareCB in the sense of Assumption 2b in Foster and Rakhlin [29].

The oracle is trained with the default VW learning rule, which performs online gradient descent with adaptive updates [27, 41, 56]. We treat the algorithm's step size parameter as a tunable hyperparameter.

For SquareCB.S, we configure SquareCB exactly as described in Foster et al. [35]. We take $\mathcal{F}$ to be the class of linear models

$$\mathcal{F} = \big\{ (x, a) \mapsto \langle w, \phi(x, a) \rangle \mid w \in \mathbb{R}^d \big\},$$

and the oracle applies the default VW learning rule to the square loss. We use the same hyperparameter range as for SquareCB.L and FastCB.L, both for the SquareCB learning rate and for the VW learning rule's step size.

**Tables in Figure 1.** For both tables, each cell $(a, b)$ plots the number of datasets in which algorithm $a$ significantly beats $b$, minus the number of datasets in which $b$ significantly beats $a$. Following Bietti et al. [13], we define a significant win using a heuristic based on an approximate $Z$-test. If $p_a$ and $p_b$ are the final PV loss values for algorithms $a$ and $b$, respectively, we say that $a$ significantly beats $b$ if

$$1 - \Phi\left( \frac{p_a - p_b}{\sqrt{\frac{p_a(1-p_a)}{n} + \frac{p_b(1-p_b)}{n}}} \right) < 0.05, \tag{41}$$

where $n$ is the number of examples and $\Phi$ is the Gauss error function.

In the left table, we choose the configuration (hyperparameters for SquareCB/FastCB and learning rate for the VW learner) with lowest final PV loss for each algorithm on a per-dataset basis. In the right table, for each algorithm we choose the hyperparameter configuration with best performance on a held-out collection of 200 datasets using the method described in Bietti et al. [13]. We keep this configuration fixed and tune only the learning rate for the VW learner on each dataset.

**Plots in Figure 1.** Each plot shows the progressive validation loss $L_{\text{PV}}(t)$ as a function of the number of examples $t$, for the best-performing (in terms of final PV loss) hyperparameter configuration for each algorithm. We consider 10 replicates for each dataset, where each replicate has the example order randomly permuted, and plot the average progressive validation loss across the replicates. Error bands

in each plot correspond to significance $p < 0.05$ under the $Z$-test in (41), setting $n = t \cdot (\#\text{replicates})$ at each time $t$.

The algorithm Supervised.L included in each of the plots is an oracle benchmark that runs online logistic regression using the true label for each example (which the bandit algorithms do not have access to). The only hyperparameter for this algorithm is the learning rate for the VW learning rule.

### E.3 Additional Figures

Figure 2 shows the results for the experiment in Figure 1 (Top-Left) with two additional adaptive algorithms, AdaCB and RegCB, included. These algorithms were found to have the strongest overall performance on the bake-off suite in Foster et al. [35] using the same online square loss oracle as SquareCB.S, and are considered state-of-the-art [13, 35]. We see in that switching SquareCB from regression with the square loss to the logistic loss (SquareCB.L) is already enough to outperform AdaCB and RegCB, and that the performance of FastCB.L is even stronger. It would be interesting to understand how the performance of AdaCB and RegCB improves if we switch to the generalized linear model (39) in the same fashion as SquareCB.L/FastCB.L, but it is unclear how to efficiently compute the confidence sets required by these algorithms in this case. We leave this for future work.

| ↓ vs → | R.S | A.S | S.S | S.L | F.L |
|---|---|---|---|---|---|
| RegCB.S | - | 6 | 46 | -6 | -12 |
| AdaCB.S | -6 | - | 42 | -8 | -18 |
| SquareCB.S | -46 | -42 | - | -55 | -66 |
| SquareCB.L | 6 | 8 | 55 | - | -11 |
| **FastCB.L** | **12** | **18** | **66** | **11** | - |

Figure 2: Head-to-head win-loss differences. Each entry indicates the statistically significant win-loss difference between the row algorithm and the column algorithm. Hyperparameters are per-dataset.