# OpenReview forum: "Efficient First-Order Contextual Bandits: Prediction, Allocation, and Triangular Discrimination"
_NeurIPS.cc/2021/Conference — NeurIPS 2021 Oral_

### Official Review · Reviewer_8uE8 · 2021-07-09

**Rating:** 7
**Confidence:** 4

**Summary:**

The authors establish first-order regret guarantees for contextual bandits, that is, the $\sqrt{T}$ dependence on the horizon is replaced by the total loss incurred by the optimal policy. This requires the algorithm to adapt to "low noise".
The proposed algorithm, called FastCB, is a variant of SquareCB by Foster et al, which is based on a reduction to online regression. These kind of algorithms have two components: an online "regression oracle" that learns to predict rewards, and an allocation rule that selects actions based on these predictions. The main innovation w.r.t. SquareCB is replacing the loss of the regression oracle, from square to logarithmic. The allocation rule is also modified to exploit this change. The authors derive first-order regret bound for FastCB.
The change of loss is further motivated by analyzing the off-line cost-sensitive classification setting, which could be of independent interest. The authors show that a plug-in classifier optimizing the squared loss cannot achieve first-order guarantees in general, while the latter is possible by using the logarithmic loss. The argument is two steps: first the excess risk is bounded by the triangular discrimination between true and predicted loss. Then they show that, by minimizing the logarithmic loss, the algorithm also makes the triangular discrimination small. The authors conjecture that triangular discrimination could be a useful tool for studying plug-in classification and contextual bandits in general.
Extensive experiments show that FastCB improves over SquareCB. Ablation shows that the gain is mostly due to the change of loss.


**Limitations And Societal Impact:**

As mentioned by the authors, the work is mostly theoretical, hence of little immediate impact.

**Main Review:**

Overall, this seems a solid contribution to the contextual bandit literature.
I checked the proofs and found no major problems.
The paper is also well written, clear, and well organized. The authors make a good job in communicating the key ideas and the logic behind the theoretical analysis.
Originality is limited, since the proposed algorithm is a variant of SquareCB. The main elements of novelty are in the theoretical methodology. The way triangular discrimination is used to derive upper bounds could indeed be a useful addition to the bandit theory tool set (however, see my issue below)
The results are nonetheless significant: in fact, this paper solves (a variant of) a COLT open problem by Agarwal et al.

My main issue/question is on the importance given to the triangular discrimination. Although it does appear in key theoretical passages, it is only to be bounded by other divergences. The key concept of Appendix B seems to be the Hellinger divergence, not the triangular discrimination (in Theorem 5 the triangular discrimination is even completely bypassed with a Hellinger bound).
In appendix C, the same happens with the KL divergence. The final algorithm has no trace of this concept, only implicitly making the triangular discrimination small.
What motivates the importance this paper assigns to the triangular discrimination, besides proof techinques? Does it have an intuitive meaning in the context of plug-in classification and contextual bandits? Can we gain something by using triangular discrimination in a more explicit way in algorithms?

Minor issues:
1. You should state up front that what you solve is a variant (albeit small) of the original COLT open problem
2. In the proof of Theorem 6, I think the key point is the following (lines 741-746): since the regressor is minimizing the log loss, and you are assuming realizability, the empirical log loss of the regressor is less or equal than that of the true function. This appears in the proof as $C_a(\widehat{f},D)\ge 0$. I think the logic behind this passage should be made more explicit since it's the point where you show that optimizing the log loss is important.
3. The appendix tends to be less organized, with ancillary results appearing suddenly in the main proofs, sometimes with in-place and sometimes with dislocated proofs. A better organization would make the proofs easier to follow.
4. Experiments: you should report the number of independent runs and the meaning of the confidence areas in the main paper too
5. I would mention that $A=|\mathcal{A}|$ at the beginning of Section 2
6. Using "regret" also for the offline setting is a bit confusing. "Excess risk" would be more clear. For the same reason, I would replace $L^\star$ with a different notation in Section 3.1
7. Line 273 "even for SquareCB": this is not very clear. Do you mean "without changing the allocation rule"?
8. Line 742: what is $y_i(a)$?
9. In the proof of Lemma 3, I think $\pi^\star$ should be $\pi$

**Time Spent Reviewing:**

8

---

> ### Author Response · Authors · 2021-08-10
> **Reviewer 8uE8**
>
> We appreciate your thorough review and positive comments! Please see responses to specific questions below.
>
> Main comments:
>
> > Overall, this seems a solid contribution to the contextual bandit literature. I checked the proofs and found no major problems. The paper is also well written, clear, and well organized. The authors make a good job in communicating the key ideas and the logic behind the theoretical analysis. Originality is limited, since the proposed algorithm is a variant of SquareCB. The main elements of novelty are in the theoretical methodology. The way triangular discrimination is used to derive upper bounds could indeed be a useful addition to the bandit theory tool set (however, see my issue below) The results are nonetheless significant: in fact, this paper solves (a variant of) a COLT open problem by Agarwal et al.
>
> **Response:** Thanks for the kind words! As a nit-pick, we would like to rebut your comment that “originality is limited.” Prior to our work, it was not known that plug-in methods can achieve first order bounds in full information, so it was not at all clear that SquareCB or any variant would work. As such, we believe our results are quite original, and have substantial technical novelty.
>
> > My main issue/question is on the importance given to the triangular discrimination. Although it does appear in key theoretical passages, it is only to be bounded by other divergences...
>
> **Response:** These are all great questions! You are completely correct that triangular discrimination plays an entirely analytical role, makes no appearance in the algorithm, and also that Theorem 5 is essentially a Hellinger bound. Indeed, triangular discrimination and Hellinger divergence are equivalent up to constants, so any bound stated in terms of one quantity can be equivalently stated in terms of the other. However, we are not aware of an approach to prove Lemma 1 without invoking the triangular discrimination as an intermediate quantity, and Lemma 1 is the key link relating the cost-sensitive risk to the regression error. The same applies to Theorem 4, which is the key result relating contextual bandit error and regression error.  Lastly, we believe that triangular discrimination has an intuitive interpretation in the cost-sensitive/CB setting, as it standardizes terms in much the same way that weighted least squares is used to combat heteroscedasticity in regression, and this is the key challenge in obtaining first-order bounds.
>
> Regarding using triangular discrimination explicitly in algorithms: We do not know if anything can be gained by this in bandits or other settings, but we hope that by raising awareness of this quantity, ML researchers may look into this in future work!
>
> ---
>
> Minor comments:
>
> > You should state up front that what you solve is a variant (albeit small) of the original COLT open problem.
>
> **Response:** We believe we are quite honest about this (cf. the paragraph “On the regression oracle model” at the bottom of page 4), but we can emphasize this more explicitly in the abstract/intro.
>
> > In the proof of Theorem 6, I think the key point is the following (lines 741-746): since the regressor is minimizing the log loss, and you are assuming realizability, the empirical log loss of the regressor is less or equal than that of the true function. This appears in the proof as
> . I think the logic behind this passage should be made more explicit since it's the point where you show that optimizing the log loss is important.
>
> **Response:** Great idea, we can definitely do this! Note that this is also a standard/crucial step in the analysis of empirical risk minimization more broadly.
>
> > The appendix tends to be less organized, with ancillary results appearing suddenly in the main proofs, sometimes with in-place and sometimes with dislocated proofs. A better organization would make the proofs easier to follow.
>
> **Response:** Thanks! We will address this in the final version.
>
> > Experiments: you should report the number of independent runs and the meaning of the confidence areas in the main paper too
>
> **Response:** We’ll try but we are quite space-limited. For reference, we did 10 independent runs for the plots and significance is measured via an approximate Z-test using the progressive validation loss (cf. Appendix E).
>
> > 5. I would mention that |\Acal| = A  at the beginning of Section 2;
> 6. Using "regret" also for the offline setting is a bit confusing. "Excess risk" would be more clear. For the same reason, I would replace  with a different notation in Section 3.1;
> 7. Line 273 "even for SquareCB": this is not very clear. Do you mean "without changing the allocation rule"?;
> 8. Line 742: what is \ell_i(a)?;
> 9. In the proof of Lemma 3, I think \pi* should be \pi.
>
> **Response:** Thanks for all of these comments/suggestions! We will make the appropriate changes. Re: line 273: Yes, we mean without changing the allocation rule. We can clarify. Re: Line 742: it should be \ell_i(a). Re: Lemma 3: yes it should be \pi. Thanks again!

---

> > ### Comment · Reviewer_8uE8 · 2021-08-16
> > **On author response**
> >
> > Thanks for the response, it provides additional insights.
> > I confirm my positive score.

---

### Official Review · Reviewer_JNTi · 2021-07-10

**Rating:** 7
**Confidence:** 3

**Summary:**

Optimal first-order regret bound for contextual bandits.

**Main Review:**

This paper built on the recent SquareCB paper and derived an optimal first-order regret bound for contextual bandits. The main change of the algorithm is to use a cross-entropy loss instead of square loss. It's interesting that the authors claim that SquareCB can not achieve the optimal first-order bound which means cross-entropy loss improves a lot. I nearly have no complaint about this work since the contribution is clear and the new technique used is highlighted. Experiments are included which is appreciated for a theory paper. One minor point is that if it is possible to design experiments to verify your conjecture or argument about the sub-optimality of SquareCB in terms of the first-order regret. Or is there any way to design experiments to justify the first-order regret is a strong measure?

======

I have read the response.

**Time Spent Reviewing:**

2

---

> ### Author Response · Authors · 2021-08-10
> **Response to Reviewer JNTi**
>
> Thanks for your positive review! We've responded to your last question below.
>
> > One minor point is that if it is possible to design experiments to verify your conjecture or argument about the sub-optimality of SquareCB in terms of the first-order regret. Or is there any way to design experiments to justify the first-order regret is a strong measure?
>
> **Response:** We certainly believe it is possible to design synthetic experiments to verify that SquareCB cannot achieve a first-order bound (indeed, we can simply generate synthetic data from the construction used to prove Theorem 2), though we feel that the experimental evaluation we have conducted with realistic data sources is more meaningful (and we are quite-space limited). Theoretically, the reasons that SquareCB fails are (a) vanilla square loss cannot adapt to heteroscedastic noise (which is a well-understood phenomenon and easy to observe experimentally; cf references in Appendix A.1) and (b) the IPW selection scheme is not aggressive enough.

---

### Official Review · Reviewer_iChz · 2021-07-13

**Rating:** 8
**Confidence:** 3

**Summary:**

This work is about contextual bandits in the low noise regime. It introduces the first efficient algorithm with a small loss bound. The latter is close to a previous algorithm SquareCB. However, the main idea is to use an online regression oracle working with a log-loss instead of a square loss. They also show that this change is fundamental as SquareCB can not achieve small loss bound. Some experiments have also been done.

**Limitations And Societal Impact:**

-

**Main Review:**

The main contribution of paper is to have design the first efficient algorithm for contextual bandits with small loss bound under the realizability assumption. The algorithm relies on a regression oracle using the log-loss. The authors show that using the log-loss is key to achieve the small loss bound.

The paper is very well written. I particularly appreciate the effort to explain the main ideas that make the proof work.
Theorem and assumptions are well stated. The proofs are easy to follow. I have only checked the core of the proof of Theorem 1 which seems correct.

The results seems important both in theory and practice. Moreover, the theoretical tools used like triangular discrimination are promising to be used in future works.

To conclude, this work is clearly a good paper.

I have only few remarks :
- It may interesting to give real computational complexity for several cases. Indeed, the efficiency of the algorithm may relativized because some regression algorithms for log-loss are not so cheap e.g. exponential forecaster.
- In example 2 (p.23), the references given suggest that the log-loss with linear function could be seen as a portfolio selection problem. However, it is not completely clear to me that it is true. For example, the loss induced by d=2, \phi(x,a) = [1,0], y =1/2 does not seem to correspond to a portfolio selection loss.

Typos/omissions:
- l.113 definition of z_t ?
- l.216 oft = often ?

**Time Spent Reviewing:**

6

---

> ### Author Response · Authors · 2021-08-10
> **Response to Reviewer iChz**
>
> Thank you for the positive comments! Please see answers to specific questions below.
>
> 1. > It may be interesting to give real computational complexity for several cases. Indeed, the efficiency of the algorithm may be relativized because some regression algorithms for log-loss are not so cheap e.g. exponential forecaster.
>
> **Response:** Good point! We have actually provided this in Appendix C.2 in the supplementary material, but unfortunately could not include it in the main body due to space constraints. In our opinion the most interesting cases are d-dimensional linear functions, where we have an end-to-end poly(d,T) time algorithm. This contrasts with the prior work, which, after discretization, would require exp(d)*T time.
>
> 2. > In example 2 (p.23), the references given suggest that the log-loss with linear function could be seen as a portfolio selection problem. However, it is not completely clear to me that it is true. For example, the loss induced by d=2, \phi(x,a) = [1,0], y =1/2 does not seem to correspond to a portfolio selection loss.
>
> **Response:** You are right that it is not exactly a portfolio selection problem. To be more precise, while our setup is not exactly a portfolio selection problem, it directly reduces to portfolio selection. When y_t is binary, we reduce by feeding in features \phi(x_t,a_t) when y_t=1, and features \mathbb{1}_d - phi(x_t,a_t) when y_t=0 (here \mathbb{1}_d is the all-ones vector). The case where y_t is in [0,1] can be reduced to this setting by sampling from Bernoulli(y_t). We will be sure to elaborate in the final version.
>
> 3. > Typos/omissions...
>
> **Response:** Thanks for catching both typos! z_t should be (x_t,a_t) and we will change oft -> often.

---

### Official Review · Reviewer_KdtT · 2021-07-17

**Rating:** 7
**Confidence:** 3

**Summary:**

The authors propose a novel algorithm for contextual bandits in the adversarial setting, where the loss functions belong to some parametric class. They show that their algorithm achieves the same "first order bound" (where the regret is upper bounded by a function of the loss of the optimal arm) as known algorithms, while being, in some cases, easier to implement. Overall the paper is well written although quite technical in some places.

**Main Review:**

1) The algorithm requires the value of the loss of the best arm as an input. How is one supposed to know this information in practice ? Knowing this information certainly is possible once all of the losses have been observed, but otherwise it seems difficult.

2) The authors discuss "computational efficiency" only by discussing how many times the prediction subroutine must be called. However, while this might be of theoretical interest (if for some reason one only wants to consider algorithms that are based on a prediction subroutine as described by the authors), what matters most is the actual amount of time and space required to implement the whole algorithm. A discussion of this and a comparison with existing algorithms would make the argument that the proposed algorithm is computationally faster more convincing.

3) The paper contains numerical experiments, which is very good. What is perhaps missing is the comparison of running times of different algorithms (since the main claim of the authors is the design of computationally efficient algorithms).

Minor Remarks:
- Isn't Assumption 2 always true providing that $Reg_{KL}(T)$ is chosen large enough ? Isn't this simply a definition ?
- In equation (3) what is $z_t$ ? And why is it that function $f$ now only takes one argument ? It was defined as a function of two arguments.
- the way that the algorithm is presented in the introduction makes it sound more complex than it actually is, and the algorithm operates as do all adversarial bandit algorithms: they predict the loss of each arm using some estimation procedure, and then draw a distribution where the probability to select an action is some simple decreasing function of the loss estimate.

**Time Spent Reviewing:**

2

---

> ### Author Response · Authors · 2021-08-10
> **Response to Reviewer KdtT**
>
> Thank you for your positive review. Please see responses to specific comments below.
>
> Major comments:
>
> 1. > The algorithm requires the value of the loss of the best arm as an input. How is one supposed to know this information in practice? Knowing this information certainly is possible once all of the losses have been observed, but otherwise it seems difficult.
>
> **Response:** Please see footnote 2, right before the statement of theorem 1 in the paper for a response to this point. The short answer is that a standard doubling trick can be used to avoid needing this information, where we set \gamma based on the learner’s observed cumulative loss \hat{L} and restart the algorithm every time this quantity doubles. The reason this trick works is that any regret bound scaling with \sqrt{\hat{L}} implies a regret bound scaling with \sqrt{L*} via a standard self-bounding argument; this is why only estimating \hat{L} is required. This is a fairly well-known technique within the cited literature, but we are happy to include a formal proof with details for completeness.
>
> 2. > The authors discuss "computational efficiency" only by discussing how many times the prediction subroutine must be called. However, while this might be of theoretical interest (if for some reason one only wants to consider algorithms that are based on a prediction subroutine as described by the authors), what matters most is the actual amount of time and space required to implement the whole algorithm. A discussion of this and a comparison with existing algorithms would make the argument that the proposed algorithm is computationally faster more convincing.
>
> **Response:**  It is somewhat standard in the literature to regard oracle-based algorithms as exponentially more efficient than non-oracle counterparts (see, e.g., Agarwal et al. (2014) for an overview). This is true in active learning, contextual bandits, and reinforcement learning more broadly. In fact, non-oracle based algorithms are almost never even implemented in practice, and are typically regarded as unimplementable for essentially all settings of interest.
>
> This pertains to our setting as well, where the only previously known first-order algorithm [Allen-Zhu et al. 2018] runs in time |\Pi|*T when invoked with a policy class |\Pi|, while for many settings of interest our algorithm can be implemented in poly(log |\Pi|, T) time (see Appendix C.2 for examples). For some end-to-end computational guarantees, we highlight two examples discussed in Appendix C.2:
> - In Example 2, we mention d-dimensional linear functions. Here the online oracle can be implemented in poly(d,T) time per step, so FastCB runs in poly(d,T) time in total and O(d) space.
> - Example 3, considers high-dimensional linear functions. Here the oracle is simply a regularized gradient descent-style update, resulting in a total running time of O(dT) and O(d) memory.
>
> In fact, no prior result formally matches these guarantees even statistically, since the only prior first-order bound [Allen-Zhu et al. 2018] considers finite policy sets. Nevertheless, a reasonable approximate instantiation of the prior work would result in \exp(d)*T running time and \exp(d) memory in both of the above settings, so our method is exponentially faster.
>
> 3. > The paper contains numerical experiments, which is very good. What is perhaps missing is the comparison of running times of different algorithms (since the main claim of the authors is the design of computationally efficient algorithms).
>
> **Response:** All algorithms considered in the experiments run in linear time over the dataset, and differences in runtime are negligible; (note that we did not implement any non-oracle based algorithms). To clear up some confusion, while our contribution is indeed computational, as discussed above, the most significant advance is the difference between “implementable” and “unimplementable” (cf. point 2 above) which we believe to be quite significant.
>
>
> ---
>
> Minor remarks:
> 1. > Isn't Assumption 2 always true providing that Reg_KL(T)  is chosen large enough ? Isn't this simply a definition ?
>
> **Response:**  You are right, as stated this is just a definition. The (implicit) assumption is that Reg_KL(T) = o(T), which is the non-trivial case. We can make this more clear.
>
> 2. > In equation (3) what is z_t? And why is it that function  now only takes one argument? It was defined as a function of two arguments.
>
> **Response:** Thanks for catching this! This is a typo. z_t should be (x_t,a_t).
>
> 3. > The way that the algorithm is presented in the introduction makes it sound more complex than it actually is, and the algorithm operates as do all adversarial bandit algorithms: they predict the loss of each arm using some estimation procedure, and then draw a distribution where the probability to select an action is some simple decreasing function of the loss estimate.
>
> **Response:** We believe it is important to emphasize the distinction between bandits and contextual bandits here. For adversarial bandits without contexts, you are correct that most algorithms can be interpreted in this way. On the other hand, most contextual bandit algorithms actually do not operate this fashion at all. Consider Exp4, the most classical adversarial contextual bandit algorithm. This algorithm uses importance weighting to estimate the loss of each policy, then then forms a distribution over policies which induces a distribution over actions. For round t, it does not seem possible to interpret the algorithm as predicting the losses for the current context x_t before the action is selected, and there is certainly no decreasing function property on the actions. Indeed, most CB algorithms operate primarily in the “policy space”, which introduces all of the computational challenges addressed by oracle-efficient algorithms. We circumvent this by operating largely in the action space.

---

> > ### Author Response · Authors · 2021-09-01
> > **Following up**
> >
> > Thanks again for your review! Please let us know if there are any further questions we can address.

---

### Decision · Program_Chairs · 2021-09-27

**Decision:**

Accept (Oral)

**Comment:**

The paper advances the state of the art in contextual bandits by proposing an interesting extension of the SquareCB algorithm of Foster and Rakhlin (ICML 2020), and showing that this method efficiently achieves optimal first-order regret guarantees under realistic conditions. This solves a COLT open problem.

The paper has received uniformly positive reviews, with the reviewers praising the solid theoretical contributions and the thorough experimental evaluation. I myself have found the contribution to be significant and the techniques to be highly original. Overall, I believe that this outstanding paper is likely to have a long-lasting impact in the contextual-bandit literature, and clearly it should be accepted for publication at NeurIPS 2021.